# Condensin controls cellular RNA levels through the accurate segregation of chromosomes instead of directly regulating transcription

Clémence Hocquet[1,2†], Xavier Robellet[1,2†], Laurent Modolo[1,2], Xi-Ming Sun[3,4], Claire Burny[1,2], Sara Cuylen-Haering[5], Esther Toselli[1,2], Sandra Clauder-Münster[6], Lars Steinmetz[6], Christian H Haering[5], Samuel Marguerat[3,4], Pascal Bernard[1,2*]

[1]CNRS Laboratory of Biology and Modelling of the Cell, Lyon, France; [2]Université de Lyon, ENSL, UCBL, Lyon, France; [3]MRC London Institute of Medical Sciences, London, United Kingdom; [4]Institute of Clinical Sciences, Faculty of Medicine, Imperial College London, London, United Kingdom; [5]Cell Biology and Biophysics Unit, Structural and Computational Biology Unit, European Molecular Biology Laboratory, Heidelberg, Germany; [6]Genome Biology Unit, European Molecular Biology Laboratory, Heidelberg, Germany

**Abstract** Condensins are genome organisers that shape chromosomes and promote their accurate transmission. Several studies have also implicated condensins in gene expression, although any mechanisms have remained enigmatic. Here, we report on the role of condensin in gene expression in fission and budding yeasts. In contrast to previous studies, we provide compelling evidence that condensin plays no direct role in the maintenance of the transcriptome, neither during interphase nor during mitosis. We further show that the changes in gene expression in post-mitotic fission yeast cells that result from condensin inactivation are largely a consequence of chromosome missegregation during anaphase, which notably depletes the RNA-exosome from daughter cells. Crucially, preventing karyotype abnormalities in daughter cells restores a normal transcriptome despite condensin inactivation. Thus, chromosome instability, rather than a direct role of condensin in the transcription process, changes gene expression. This knowledge challenges the concept of gene regulation by canonical condensin complexes.
DOI: https://doi.org/10.7554/eLife.38517.001

*For correspondence:
pascal.bernard@ens-lyon.fr

†These authors contributed equally to this work

Competing interests: The authors declare that no competing interests exist.

## Introduction

Structural Maintenance of Chromosomes (SMC) complexes are ring-shaped ATPases, conserved from bacteria to human, which shape chromosomes and ensure their accurate transmission during cell divisions (*Hirano, 2016*; *Uhlmann, 2016*). Eukaryotes possess three distinct SMC protein complexes, named condensins, cohesin and SMC5/6. Condensins structure and condense chromosomes, cohesin mediates sister-chromatid cohesion and organises topological domains in the genome during interphase, and SMC5/6 promotes proper DNA replication and repair (*Hirano, 2016*; *Uhlmann, 2016*). A large body of in vivo and in vitro studies has substantiated the idea that SMC complexes shape the genome and preserve its integrity by encircling DNA helixes and, at least partly, by extruding loops of DNA (*Ganji et al., 2018*; *Gibcus et al., 2018*; *Hirano, 2016*; *Uhlmann, 2016*; *van Ruiten and Rowland, 2018*). Besides organising chromosomes, cohesin and condensins have also been widely implicated in the control of gene expression, raising the idea that the two complexes link gene expression to chromosome architecture (*Dowen and Young, 2014*). Yet,

while our understanding of gene regulation by cohesin has progressed during the last decade (*Merkenschlager and Nora, 2016*), the mechanisms through which condensins impact on gene expression have remained largely enigmatic.

Condensins have been best characterised as the key drivers of the assembly of mitotic chromosomes (*Gibcus et al., 2018*; *Hirano, 2016*; *Robellet et al., 2017*). The profound reorganisation of chromatin fibres into compact and individualised rod-shaped chromosomes that marks the entry into mitosis is essential for the accurate transmission of the genome during anaphase. When the function of condensins is impaired, chromosome arms remain entangled and hence fail to separate, leading to the formation of sustained anaphase chromatin bridges and DNA breakage during telophase and in post-mitotic cells (*Cuylen et al., 2013*; *Kim et al., 2009*; *Samoshkin et al., 2012*; *Sutani et al., 1999*; *Toselli-Mollereau et al., 2016*; *Woodward et al., 2016*).

Like all SMC complexes, condensins are composed of two ATPases, called SMC2$^{Cut14}$ and SMC4$^{Cut3}$ (fission yeast names are indicated in superscript), that associate with three non-SMC subunits, which regulate the ATPase activity of the holocomplex and govern its association with DNA (*Kschonsak et al., 2017*). Most eukaryotes possess two condensins, called condensin I and II, which are composed of the same SMC2/SMC4 heterodimer but two different sets of non-SMC subunits (*Hirano, 2016*; *Robellet et al., 2017*). Despite their structural and functional similarities, the dynamics of association of condensin I and II with chromosomes differ (*Walther et al., 2018*). Condensin II is nuclear during interphase and enriched on chromosomes from prophase until telophase. Condensin I, in contrast, is mostly cytoplasmic during interphase and associates with chromosomes from prometaphase until telophase. A third condensin variant called the Dosage Compensation Complex (DCC) has been described in the worm *Caenorhabditis elegans*, which associates with the two X chromosomes in hermaphrodite animals and halves the expression of X-linked genes by reducing the occupancy of RNA polymerase II (RNA Pol II) (*Kruesi et al., 2013*). Yeasts, in contrast, possess a unique condensin complex, similar in terms of primary sequence to condensin I.

Condensin I and II have been implicated in the control of gene expression in various organisms, ranging from yeasts to human. In the budding yeast *Saccharomyces cerevisiae*, the silencing of heterochromatic mating type genes and the position effect exerted by repetitive ribosomal DNA are both alleviated when condensin is impaired (*Bhalla et al., 2002*; *Wang et al., 2016*). Likewise, in the fission yeast *Schizosaccharomyces pombe*, the SMC4$^{Cut3}$ subunit of condensin is needed to repress tRNAs genes as well as reporter genes inserted into the pericentric DNA repeats, which are coated with heterochomatin (*He et al., 2016*; *Iwasaki et al., 2010*). It remains unclear, however, during which cell cycle phase and through which mechanism budding and fission yeast condensins regulate the expression of such diverse genes in such different genomic contexts. In *C. elegans*, depletion of condensin II is linked to an increase in the expression of at least 250 genes, but this effect does not correlate with the occupancy of condensin II on chromosomes (*Kranz et al., 2013*). In *Drosophila melanogaster*, condensin I has been implicated in the repression of homeotic genes (*Lupo et al., 2001*), whereas condensin II has been reported to take part in the production of antimicrobial peptides (*Longworth et al., 2012*). Murine peripheral T cells that express a mutant version of the condensin II CAP-H2 regulatory subunit, exhibit a decreased compaction of chromatin and an increased expression of the proliferative gene *Cis* (*Rawlings et al., 2011*), suggesting a possible mechanistic relationship between condensin-mediated chromosome organisation and gene expression. However, another study reported that the same CAP-H2 mutation led to only subtle effects on the transcriptome of precursor thymocytes, which might be caused by chromosomal instability (*Woodward et al., 2016*).

Perhaps more consistent with a direct role in gene regulation, condensin I was found associated with active promoters during M phase in chicken DT40 cells, and its depletion prior to mitotic entry coincided with a decreased expression during the subsequent G1 phase of a subset of genes to which it bound to (*Kim et al., 2013*). Similarly, cohesin and condensin II have been detected at super-enhancers in rapidly proliferating mouse embryonic stem cells, and depleting condensin II has been associated with a reduced expression of cell-identity genes driven by these super-enhancers (*Dowen et al., 2013*). Yuen et al. observed a similar effect on the expression of highly-expressed housekeeping genes in mouse embryonic stem cells and human embryonic kidney cells (*Yuen et al., 2017*). Finally, it has been reported that not only condensin II, but also condensin I, binds enhancers activated by β-estradiol during interphase in human MCF7 breast adenocarcinoma cells, and that the depletion of condensin I or II led to a reduced transcription of oestrogen-activated genes

(*Li et al., 2015*). Intriguingly, the same enhancers where also found occupied by cohesin and to rely upon this complex to drive gene expression (*Li et al., 2013*).

All these studies tend to support the idea that condensin I and II play an important and evolutionarily conserved role in gene expression, through which they impinge on cell identity, cell proliferation and, possibly, also immunity. Yet, no conclusive evidence has been provided thus far as to how condensin I and II might achieve this function. Mitotic chromosomes conserve considerable chromatin accessibility, similar to interphase chromatin (*Hihara et al., 2012*), and DNA remains accessible to transcription factors even in mitotic chromosomes that have been structured by condensin complexes (*Chen et al., 2005*; *Palozola et al., 2017*). Thus, it remains unclear whether and how mechanisms related to chromosome condensation could possibly underlie condensin-mediated gene regulation. Furthermore, given that the loss or gain of chromosomes is sufficient to alter gene expression (*Santaguida and Amon, 2015*; *Sheltzer et al., 2012*), it is crucial to determine to which extent the role attributed to condensin I and II in the control of gene expression is mechanistically different from, or related to, the assembly and segregation of chromosomes during mitosis.

Gene expression can be controlled at the transcriptional level by changing the activity and/or the occupancy of RNA polymerases, as exemplified by condensin$^{DCC}$ (*Kruesi et al., 2013*). It can also be controlled at the co- or post-transcriptional level by modulating the half-life of transcripts (*Bühler et al., 2007*; *Harigaya et al., 2006*). The RNA-exosome is a conserved ribonuclease complex that ensures the maturation and the controlled degradation of a plethora of RNAs in the cell, including, for example, defective RNAs or cryptic unstable non-coding RNAs (*Kilchert et al., 2016*). The RNA-exosome consists of nine core subunits that associate with the RNase Dis3 in the cytoplasm, plus a second RNase, called Rrp6, in the nucleus (*Kilchert et al., 2016*). The mechanisms behind target recognition and processing or degradation modes by the RNA-exosome are not fully understood. Cofactors, such as the TRAMP poly(A)polymerase complex, stimulate the RNase activity of the RNA-exosome and specify its targets (*Kilchert et al., 2016*). Rrp6 has been found associated with chromosomes at actively transcribed genes (*Andrulis et al., 2002*), leading to the idea that the RNA-exosome can handle nuclear RNA in a co-transcriptional manner. By processing and eliminating cellular transcripts, the RNA-exosome plays a central role in proper gene expression (*Kilchert et al., 2016*).

To gain insights into how canonical condensins regulate genes, we investigated the role of condensin complexes in gene expression during the fission and budding yeast cell cycle. In contrast to previous studies, we present here converging evidence that condensin plays no major direct role in the control of gene expression, neither in *S. pombe* nor in *S. cerevisiae*. We show that lack of condensin activity is associated with increased levels of tRNAs, mRNAs and non-coding RNAs in post-mitotic fission yeast cells, reminiscent of other organisms, but that this effect is indirect: RNAs accumulate in condensin mutant cells because of the missegregation of chromosomes, as illustrated by the non-disjunction of the rDNA during anaphase, which dampens RNA degradation by the nucleolar RNA-exosome in post-mitotic cells. The discovery that budding and fission yeast condensins contribute to proper gene expression by maintaining chromosome stability during cell divisions, and not through a direct impact on gene transcription, challenges the widespread idea that condensin I and condensin II are direct regulators of gene expression.

## Results

### RNA-exosome-sensitive transcripts accumulate when condensin is impaired in fission yeast

To assess the role of condensin in gene expression in fission yeast, we compared the transcriptomes of wild-type and mutant cells in which the SMC2$^{Cut14}$ ATPase subunit of condensin was inactivated by the thermosensitive *cut14-208* mutation (*Saka et al., 1994*). We shifted cells growing asynchronously at permissive temperature to the restrictive temperature of 36°C for 2.5 hr (one cell doubling) to inactivate SMC2$^{Cut14}$ and determined their transcriptomes using strand-specific RNA-seq. We identified 306 transcripts that were differentially expressed with log2 fold changes superior to 0.5 or inferior to −0.5 (FDR ≤ 0.05) in the *cut14-208* condensin mutant compared to wild type (*Figure 1A*). The vast majority of transcripts (98.5%; n = 302/306) exhibited an increased steady-state level in the mutant. We confirmed the increase for six example RNAs by RT-qPCR, using either *act1 or nda2*

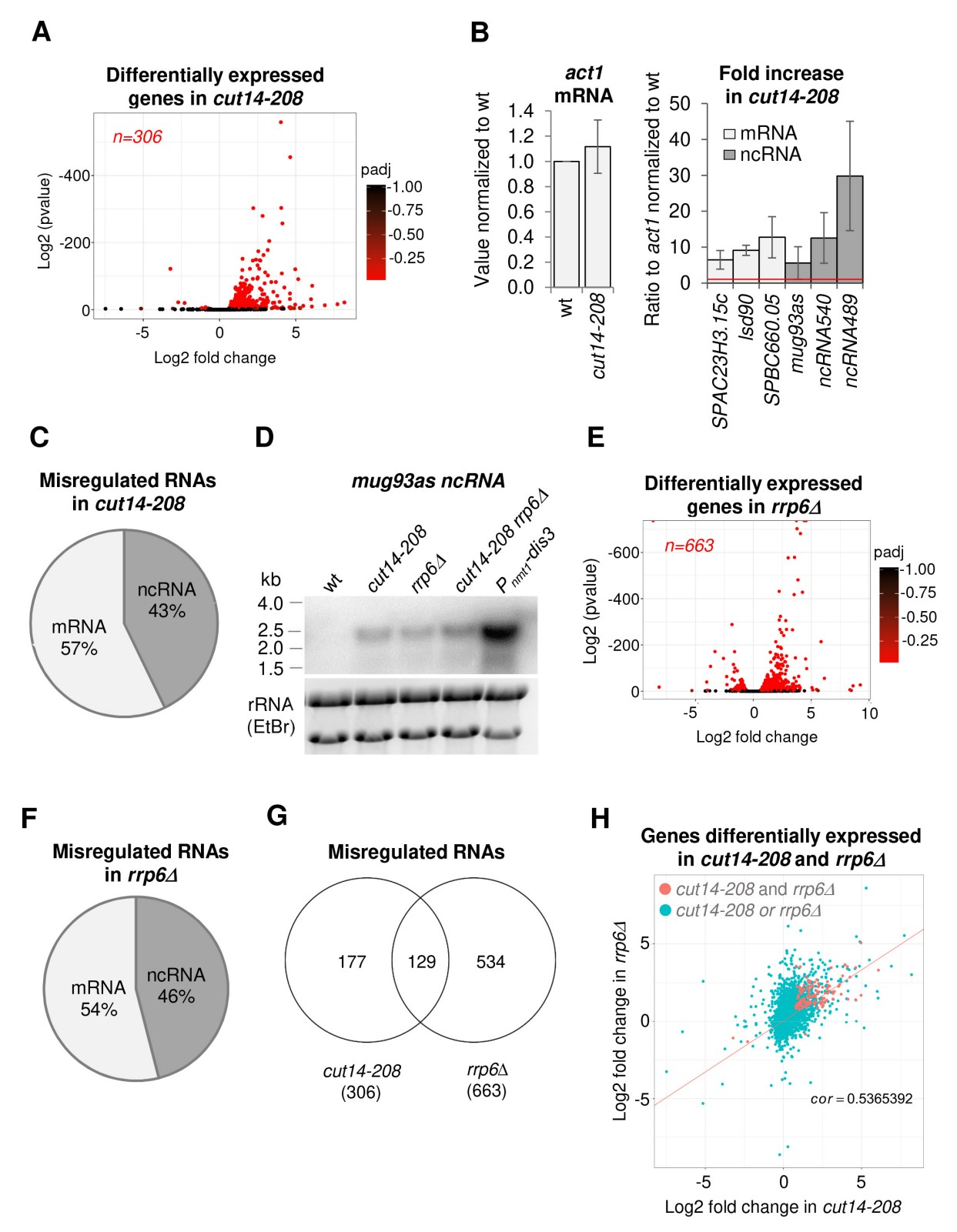

**Figure 1.** The condensin loss-of-function mutant *cut14-208* accumulates RNA-exosome-sensitive transcripts. (**A**) Volcano plot of RNA levels measured by strand-specific RNA-seq in the *cut14-208* condensin mutant after 1 cell doubling at 36°C, from biological triplicates. Genes exhibiting a log2 fold change superior to 0.5 or inferior to −0.5 with an adjusted P-value (padj) ≤ 0.05 are indicated in red. (**B**) RT-qPCR validation. Total RNA from cells grown at 36°C for 2.5 hr was reverse-transcribed in the presence or absence of Reverse Transcriptase (RT) and cDNAs were quantified by qPCR. Shown

*Figure 1 continued on next page*

Figure 1 continued

are the averages and standard deviations (SDs) measured from three biological replicates. (C) Misregulated RNA in *cut14-208*. (D) Northern blot analysis of the non-coding RNA *mug93as*. Cells were shifted at 36°C for 1 cell doubling and total RNA probed for *mug93as* level. Ribosomal RNA (rRNA) stained with ethidium bromide (EtBr) was used as loading control. (E) Volcano plot of RNA levels measured by RNA-seq from biological triplicates of the *rrp6Δ* mutant after 1 cell doubling at 36°C. Genes exhibiting a log2 fold change superior to 0.5 or inferior to −0.5 with an adjusted P-value (padj) ≤ 0.05 are indicated in red. (F) Misregulated RNA in *rrp6Δ*.( G) Genes misregulated in *cut14-208* and *rrp6Δ*. (H) Comparison plots between the transcriptomes of *cut14-208* and *rrp6Δ*. Genes differentially expressed in both mutants are highlighted in red. The correlation coefficient has been calculated for all genes.

DOI: https://doi.org/10.7554/eLife.38517.002

The following figure supplement is available for figure 1:

**Figure supplement 1.** The condensin loss-of-function mutant *cut14-208* accumulates exosome-sensitive transcripts.

DOI: https://doi.org/10.7554/eLife.38517.003

mRNA levels as internal controls (*Figure 1B*, *Figure 1—figure supplement 1A*). Thus, gene expression is altered, and mainly increased, in dividing condensin mutant *cut14-208* cells.

Of the 306 misregulated transcripts, 57% were mRNAs and the remaining 43% were non-protein coding RNAs (ncRNA) (*Figure 1C*). We found histone mRNAs amongst the upregulated transcripts, confirming previous observations (*Kim et al., 2016*). However, the analysis of all increased mRNAs revealed no enrichment for a specific gene ontology (GO) term, which suggests that condensin inactivation does not affect the expression of a particular family of protein-coding genes. In contrast, ncRNAs, which represent ~22% of the transcription units in the fission yeast genome (n = 1524/6986; genome version ASM294v2.30), were significantly enriched in the population of transcripts upregulated in the *cut14-208* mutant (p<0.001, Chi-square test). Since most ncRNAs are maintained at a low level by the nuclear RNA-exosome (*Wilhelm et al., 2008*), their controlled degradation might be compromised in the mutant. We tested this hypothesis by northern blotting using the antisense ncRNA *mug93-antisense-1* (*mug93as*) as a representative example. As shown in *Figure 1D*, *mug93as* was barely detectable in a wild-type background but accumulated in cells lacking Rrp6 or defective for Dis3, as expected if it were degraded by the RNA-exosome. Remarkably, *mug93as* also accumulated in *cut14-208* mutant cells, reaching levels reminiscent of the *rrp6Δ* control. Furthermore, chromatin immunoprecipitation (ChIP) against RNA Pol II revealed no change in RNA Pol II occupancy at neither the *mug93as* gene, nor two additional example ncRNA genes (*ncRNA.489* and *ncRNA.540*), in *cut14-208* cells (*Figure 1—figure supplement 1B*), although their ncRNA levels were increased between 5- and 30-fold (*Figure 1B*, *Figure 1—figure supplement 1A*). These RNAs might therefore accumulate due to impaired degradation rather than increased transcription. Taken together, these results indicate that unstable RNA species accumulate when condensin is defective.

To clearly delineate the number of transcripts targeted by the RNA-exosome that accumulate in the *cut14-208* condensin mutant, we compared the transcriptomes of *cut14-208* and *rrp6Δ* cells that had been grown in parallel and processed simultaneously for strand-specific RNA-seq analyses. We identified 663 RNAs that were differentially expressed with log2 fold changes superior to 0.5 or inferior to −0.5 (FDR ≤ 0.05) in the *rrp6Δ* mutant (*Figure 1E*). Of these differentially expressed RNAs, 78% were increased in levels and 22% were decreased. The population of Rrp6-sensitive RNAs was considerably enriched in ncRNAs (p<0.001, Chi-square test) (*Figure 1F*). Pairwise comparison with *cut14-208* revealed that ~ 43% (n = 129/302) of the RNAs that accumulated in the condensin mutant also accumulated in cells lacking Rrp6 (*Figure 1G,H*), with no clear preference for ncRNA and mRNA transcripts (50% each). A hypergeometric test confirmed that this overlap was statistically highly significant (p=$4.6e^{-55}$). These data indicate that a large fraction of the transcripts that accumulate when condensin is impaired are normally targeted by the ribonuclease Rrp6.

## Read-through transcripts accumulate upon condensin inactivation

Visual inspection of the RNA-seq profile of the *cut14-208* mutant revealed a widespread increase of reads downstream of the 3′ ends of genes, suggesting defects in the termination of transcription (for an example, see the *hsp9* gene in *Figure 2A*). Read-through transcripts are abnormal RNAs that are extended at their 3′ ends when RNA polymerases transcribe over Transcription Termination Sites (TTS) into downstream DNA sequences. Dis3 and Rrp6 have been reported to prevent the accumulation of read-through transcripts, although through possibly different mechanisms (*Lemay et al.,*

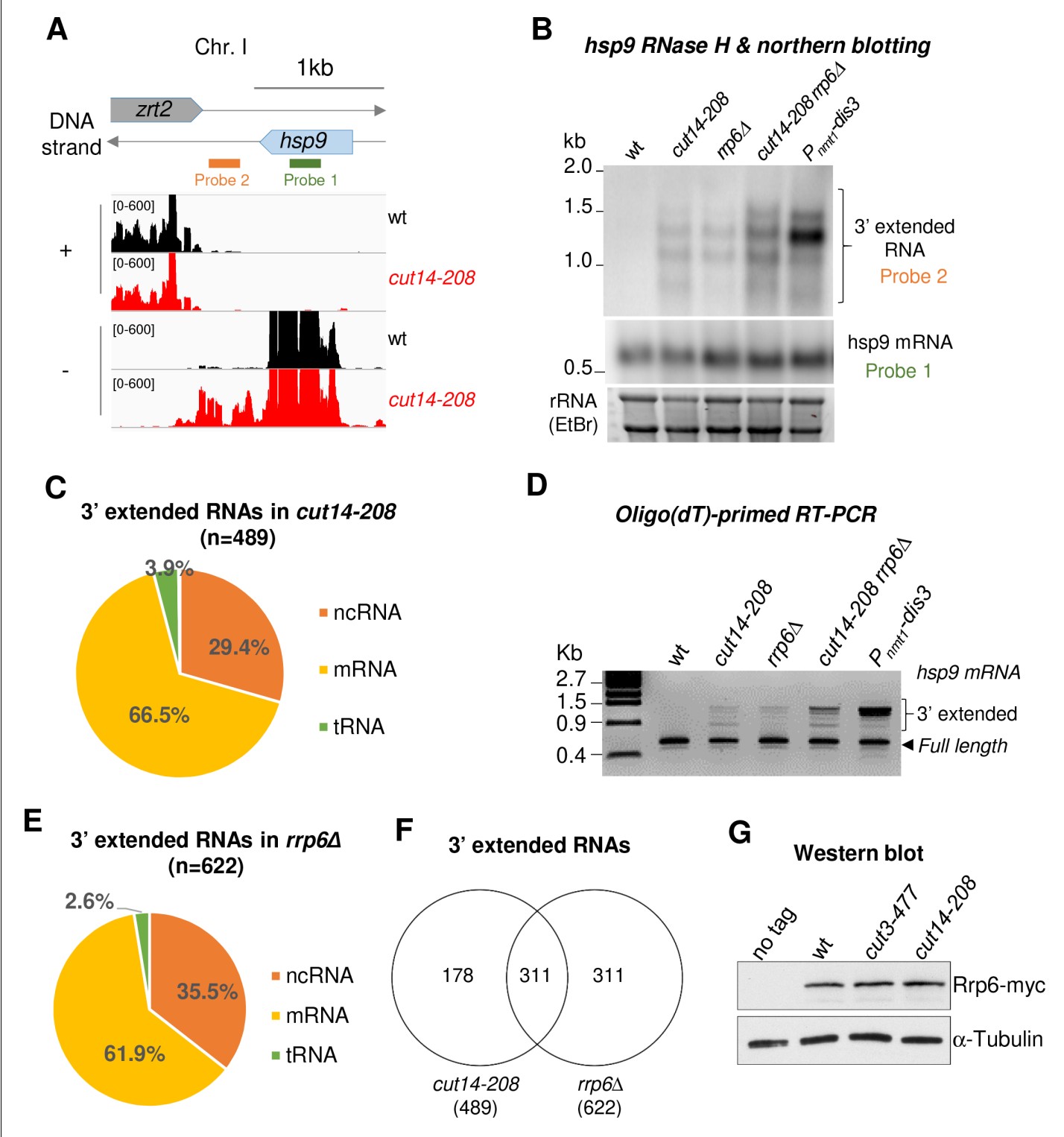

**Figure 2.** The condensin loss-of-function mutant *cut14-208* accumulates 3'-extended read-through transcripts. (**A**) 3'-extended *hsp9* read-through RNA detected by strand-specific RNA-seq in *cut14-208*. (**B**) Read-through *hsp9* RNA detected by RNase H digestion and northern blotting. Total RNA from indicated strains grown at 36°C was digested by RNase H in the presence of a DNA oligonucleotide complementary to the 5'end of *hsp9* mRNA. Cleaved products were revealed by a probe hybridizing downstream the transcription termination site of *hsp9* (see probe2 in A), or within the coding sequence (probe 1, shown in A). rDNA stained with EtBr served as loading control. (**C**) 3'-extended read-through RNAs in *cut14-208*. (**D**) Polyadenylated RNAs detected by oligo(dT)-primed RT-PCR. Cells were grown at 36°C for 2.5 hr in PMG supplemented with 60 μM thiamine to repress *nmt1-dis3*. Total

*Figure 2 continued on next page*

*Figure 2 continued*

RNA was reverse transcribed using oligo(dT) primers in the presence or absence of RT. cDNA were amplified by 25 cycles of PCR using oligo(dT) and gene specific primers. Minus RT reactions produced no signal. (E) 3'-extended read-through RNAs in *rrp6Δ*. (F) Overlap between the sets of read-through RNAs in *cut14-208* and *rrp6Δ*. (G) Steady state level of Rrp6. Indicated strains were grown at 36°C for 2.5 hr, total proteins were extracted and probed with an anti-myc antibody. Alpha-tubulin served as loading control.

DOI: https://doi.org/10.7554/eLife.38517.004

The following figure supplements are available for figure 2:

**Figure supplement 1.** The condensin loss-of-function mutant *cut14-208* accumulates 3'-extended read-through transcripts.

DOI: https://doi.org/10.7554/eLife.38517.005

**Figure supplement 2.** The condensin loss-of-function mutant *cut14-208* accumulates 3'-extended read-through transcripts.

DOI: https://doi.org/10.7554/eLife.38517.006

*2014*; *Zofall et al., 2009*). Lemay et al. have shown that 3'-extended *hsp9* RNAs accumulate in fission yeast cells when Dis3 is impaired, and to a lesser extent when Rrp6 is lacking (*Lemay et al., 2014*). By using RNase-H digestion of the 5' end of *hps9* mRNA and northern blotting, we confirmed the accumulation of 3'-extended *hsp9* RNA in *dis3* and *rrp6* mutant cells (*Figure 2B*). Importantly, *cut14-208* mutant cells accumulated 3'-extended *hsp9* RNA of the same length as cells depleted of Dis3 or Rrp6. Moreover, other condensin mutants, such as *cut3-477* and *cut14-180*, also accumulated read-through *hsp9* RNA (*Figure 2—figure supplement 1A*), which demonstrates that the accumulation of uncleaved, 3'-extended *hsp9* transcripts is a feature of condensin inactivation.

To determine the prevalence of read-through RNAs in *cut14-208* mutant cells, we systematically searched and quantified stretches of consecutive RNA-seq reads that mapped immediately downstream the TTS of annotated genes on the same DNA strand, and did not overlap with downstream genes (see Materials and Methods) (*Modolo, 2018*). Using these criteria, we identified 489 transcripts, mostly mRNAs and ncRNAs, which were extended at their 3' ends (*Figure 2C*). Oligo(dT)-primed RT-PCR showed that read-through RNAs were polyadenylated (*Figure 2D*, *Figure 2—figure supplement 1B*). Non-canonical polyadenylation sites might be more frequently used in the *cut14-208* mutant than in wild-type cells, as already described for cells lacking Dis3 (*Lemay et al., 2014*), or 3'-extended RNAs that end at non-canonical sites might be stabilised in the condensin mutant. GO term analysis revealed no specific feature defining the population of read-through transcripts that accumulate in *cut14-208* cells. Furthermore, only 111 of the 489 read-through transcripts were also up-regulated (*Figure 2—figure supplement 1C*), which suggests that the 3'-extended RNAs were not the by-product of increased transcription. In addition, ChIP against transcriptionally active RNA Pol II revealed no increased occupancy downstream canonical TTS in *cut14-208* mutant cells (*Figure 2—figure supplement 1D*), in contrast to the transcription termination mutant *pfs2-11* that we used as control (*Wang et al., 2005*). This result suggests that the termination of transcription remains largely effective in *cut14-208* mutant cells. The 3'-extended transcripts that accumulate in *cut14-208* cells are, therefore, unlikely to stem from an increased transcription beyond the TTS.

Condensin and cohesin frequently, but not systematically, co-localise along chromosomes (*Kim et al., 2016*; *Nakazawa et al., 2015*; *Schmidt et al., 2009*). It has been proposed that the accumulation of cohesin at the 3' ends of co-transcribed convergent genes locally strengthens the termination of transcription (*Gullerova and Proudfoot, 2008*). We analysed the orientation of genes that exhibited read-through transcripts in *cut14-208* condensin mutant cells, taking into account all transcription units (i.e. coding and non-coding). As previously observed (*Schmidt et al., 2009*), we found an enrichment for genes in a convergent orientation throughout the fission yeast genome (~62%, n = 4302/6986). In contrast, the read-through transcripts that accumulated in the *cut14-208* mutant emanated mostly from genes oriented in tandem (53%, n = 259/489) and slightly less from convergent genes (47%, n = 230/489). We also analysed the localisation of cohesin in *cut14-208* cells grown at the restrictive temperature, using ChIP against the SMC3$^{Psm3}$ subunit of cohesin tagged with GFP. We observed no change in the localisation of Psm3-GFP neither at genes with read-through transcripts nor at up-regulated genes in *cut14-28* mutant cells (*Figure 2—figure supplement 2A*). Read-through transcripts have been detected at the *nmt2*, *tea3*, *mei4* and *pdt1* genes in cohesin mutant cells (*Gullerova and Proudfoot, 2008*). In sharp contrast, we observed no read-through RNAs at any of these four genes in a *cut14-208* background (*Figure 2—figure supplement 2B*). Taken together, these data suggest that both cohesin binding to chromatin and cohesin-

dependent transcription termination remain largely unaffected when condensin is impaired by the *cut14-208* mutation.

To further investigate the molecular origin of read-through RNAs in condensin mutant cells, we compared *cut14-208* with *rrp6Δ* cells. We detected 622 read-through RNAs, again mostly mRNAs and ncRNAs, in the transcriptome of cells lacking Rrp6 (*Figure 2E*), which confirms previous reports (*Lemay et al., 2014*; *Zofall et al., 2009*). Pol II occupancy remained unchanged downstream the TTS of genes with read-through transcripts in *rrp6Δ* cells (*Figure 2—figure supplement 1D*), consistent with the idea that Rrp6 is needed to eliminate read-though RNAs at the co- or post-transcriptional level (*Zofall et al., 2009*). Importantly, 50% of the read-through RNAs that accumulated in cells lacking Rrp6 were also extended at their 3'ends in the *cut14-208* mutant (*Figure 2F*). This reinforces the idea that the function of Rrp6 might be affected in a *cut14-208* background. The steady state level of the Rrp6 protein remained unchanged in *cut14-208* mutant cells (*Figure 2G*). Likewise, we observed no change by RNA-seq in the mRNA levels of RNA-exosome or TRAMP components in the condensin mutant (*Supplementary file 1*). Collectively, these data indicate that 3'-extended read-through transcripts accumulate upon condensin inactivation in fission yeast and that this accumulation might stem from defects in the processing of these transcripts by Rrp6.

## Condensin is dispensable for gene expression during interphase and metaphase in fission and budding yeasts

Since condensin is regulated over the course of the cell cycle, we sought to determine the phase(s) during which condensin function is required for proper gene expression. We synchronised *cut14-208* mutant cells in early S phase at the permissive temperature, raised the temperature to 36°C to inactivate condensin and at the same time released them into the cell cycle. We then measured gene expression by RT-qPCR as cells progressed from early S phase into the cell cycle (*Figure 3A*). To ensure that cells went only through a single cell cycle at 36°C, we re-arrested cells in late G1 phase by the thermosensitive mutation *cdc10-129*. Previous work had shown that 10 min at 36°C are sufficient to inactivate condensin in *cut14-208* mutant cells (*Nakazawa et al., 2011*). FACScan analysis of DNA content and cytological observations revealed that *cdc10-129* single mutant and *cdc10-129 cut14-208* double mutant cells completed S phase (t = 30 min after release) and progressed through G2 phase (t = 60 min) and mitosis (t = 90 min) with similar kinetics (*Figure 3B*). Chromosome segregation was impaired in the *cut14-208* mutant background, as revealed by the appearance of anaphase chromatin bridges (*Figure 3B*, green line), which were subsequently severed by the septum upon mitotic exit, producing the CUT phenotype (Cells Untimely Torn) (*Figure 3B*, red line). Accordingly, FACScan analysis revealed the appearance of aberrant karyotypes in post-mitotic cells (t = 120 and 180 min; *Figure 3B*).

Remarkably, we detected no up-regulation of any of the three reporter RNAs that we had selected from the pool of upregulated transcripts in *cut14-208* mutants (*Figure 1B*) during G2 phase in synchronized *cdc10-129 cut14-208* cells (t = 60 min, *Figure 3C*). These RNAs started to accumulate, however, at t = 90 min and further increased at t = 120 min and 180 min (*Figure 3C*), coincidently with the accumulation of aneuploid post-mitotic cells. This result suggests that condensin is possibly required for proper gene expression during late mitosis or early G1 phase. To validate these results, we repeated the RT-qPCR measurements by synchronously releasing *cut14-208* cells from the early S phase block only 1 hr after shifting the temperature to 36°C and re-arrested them already at the G2/M transition using a *cdc25-22* mutation (*Figure 3D,E*). In this time course experiment, we observed no up-regulation of the reporter RNAs at any time point (*Figure 3F*). Similarly, RNA levels remained unchanged after a sustained arrest in metaphase at 36°C by depletion of the anaphase promoting complex (APC/C) activator Slp1 (*Figure 3G–I*). We conclude that, in fission yeast, the function of condensin is dispensable for gene regulation during S and G2 phases of the cell cycle, consistent with the finding that condensin is largely displaced from the nucleus during this time (*Sutani et al., 1999*).

Condensin in the budding yeast *S. cerevisiae* remains, in contrast, bound to chromosomes throughout the cell cycle (*D'Ambrosio et al., 2008b*), which is reminiscent to the continuous nuclear localization of the condensin II complex in metazoan cells. This raises the possibility that condensin controls interphase gene expression in this species. We first confirmed that condensin localizes to chromosomes of cells released synchronously into the cell cycle from a G1 mating pheromone arrest (*Figure 4—figure supplement 1A*). Chromosome spreading and ChIP showed that condensin

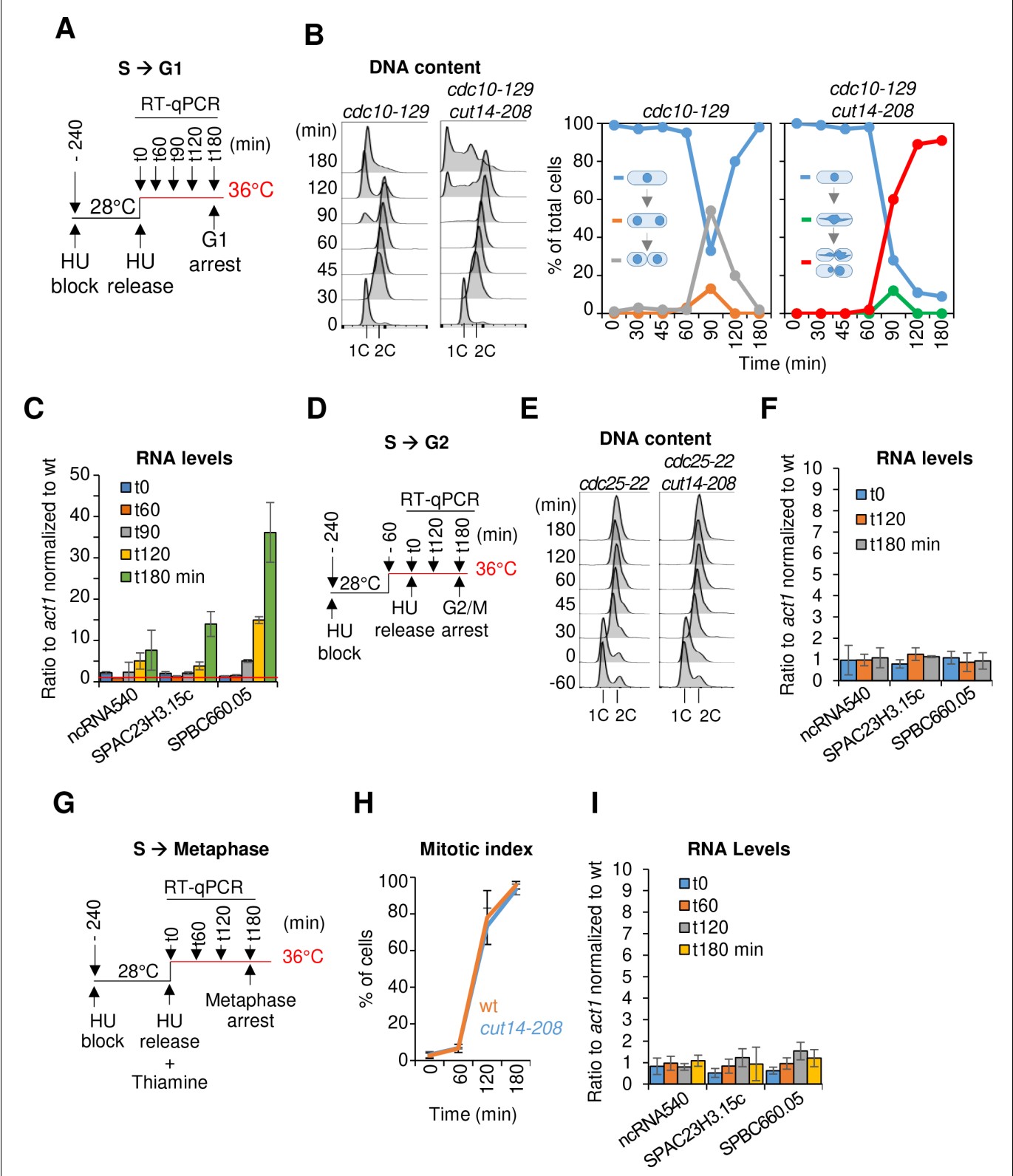

**Figure 3.** The function of condensin is dispensable for gene regulation during S and G2 phases of the cell cycle in fission yeast. (A–C) Gene expression was assessed in synchronized *cut14-208* cells progressing from early S phase to G1 phase at the restrictive temperature. (A) Scheme of the experiment. (B) Left panel: FACScan analyses. Right panels: chromosome segregation and cytokinesis assessed by staining DNA with DAPI and the septum with calcofluor (n >= 100). (C) Total RNA extracted from *cdc10-129* and *cdc10-129 cut14-208* cells shown in (B), was reverse-transcribed in the presence or

*Figure 3 continued on next page*

*Figure 3 continued*

absence of RT and cDNA quantified by qPCR. Red line = 1. Shown are averages ± SDs measured from biological triplicates. (**D–F**) Gene expression was assessed in synchronized *cut14-208* cells progressing from early S phase to late G2 phase at the restrictive temperature. (**D**) Scheme of the experiment. (**E**). FACScan analyses. (**F**) Total RNA extracted from *cdc25-22* and *cdc25-22 cut14-208* cells shown in (**E**) was reverse-transcribed in the presence or absence of RT and cDNA quantified by qPCR. Shown are averages ± SDs measured from biological triplicates. (**G–I**) Gene expression was assessed in synchronized *cut14-208* cells progressing from early S phase to metaphase at the restrictive temperature. (**G**) Scheme of the experiment. Thiamine repressed the *nmt41-slp1* gene in order to arrest cells in metaphase. (**H**) Percentages of mononucleate, mitotic cells from n = 3 experiments. (**I**) Total RNA extracted from *nmt41-slp1 and nmt41-slp1 cut14-208* cells shown in (**H**) was reverse-transcribed in the presence or absence of RT and cDNA quantified by qPCR. Shown are averages ± SDs measured from biological triplicates.
DOI: https://doi.org/10.7554/eLife.38517.007

bound to chromosomes already during G1 phase and that its levels on chromosomes increased as cells passed through S and G2 phases (*Figure 4—figure supplement 1B,C*). To inactivate condensin in budding yeast, we proteolytically cleaved the kleisin (Brn1) subunit of the condensin ring by inducing expression of a site-specific protease from tobacco etch virus (TEV) using a galactose-inducible promoter, which efficiently released condensin from chromosomes (*Cuylen et al., 2011*), even during G1 phase (*Figure 4—figure supplement 1D*). We then compared the transcriptome of G1 phase-arrested cells after condensin cleavage to cells with intact condensin (*Figure 4A*). Remarkably, solely 26 transcripts were differentially expressed by at least two-fold (*Figure 4B*). To rule out that such minor effect on gene expression was not an artefact caused by the G1 phase arrest, we repeated the experiment, but this time released cells after Brn1 TEV cleavage from the G1 phase arrest and re-arrested them in the subsequent M phase by addition of the spindle poison nocodazole, before preparing RNA for transcriptome analysis (*Figure 4C*). In this experiment, only six genes displayed an up- or downregulation of two-fold or more (*Figure 4D*). We conclude that condensin inactivation by releasing the complex from chromosomes has no major effects on the gene expression program of budding yeast cells when these cells are prevented from undergoing division.

## Gene expression changes in fission yeast condensin mutants are the result of a loss of genome integrity

The fact that we observed changes in transcript levels in asynchronously dividing *cut14-208* fission yeast mutant cells, but not in fission or budding yeast condensin mutants that are prevented from undergoing anaphase, raises the possibility that defects in chromosome segregation caused by condensin inactivation might be causally responsible for condensin-dependent gene deregulation. Indeed, when we arrested fission yeast *cut14-208* cells at the G2/M transition using the analogue-sensitive Cdc2asM17 kinase (*Aoi et al., 2014*), shifted the temperature to 36°C and then released them from the arrest to complete mitosis, we measured an increase in transcript levels after mitotic exit and the severing of chromosomes (*Figure 5—figure supplement 1A–C*).

If RNA misregulation were indeed caused by the severing of missegregated chromosomes by the cytokinetic ring, then three key hypotheses should prove correct: (1) any mutation that results in chromosome missegregation and cutting upon mitotic exit should result in an increase in levels of the same or a similar set of RNAs as the *cut14-208* mutation, (2) the amplitude of the increase in RNA levels should correlate with the prevalence of missegregation, and (3) preventing chromosome cutting in *cut14-208* cells should attenuate the changes in RNA levels. As shown in *Figure 5A*, mutation of separase (*ptr4-1*) or of a subunit of the APC/C (*cut9-665*) resulted in chromosome missegregation and severing by the cytokinetic ring upon mitotic exit, as well as an increase in RNA levels, in a manner similar to *cut14-208*. Moreover, the amplitude of the increase in RNA levels correlated with the frequency of chromosome cutting in the different mutants (*Figure 5A*). The analysis of five additional condensin mutations of increasing prevalence of chromosome missegregation further confirmed this correlation (*Figure 5—figure supplement 1D,E*).

Furthermore, we found that RNA levels remained comparable to wild-type in *cut14-208* cells that were prevented from undergoing cytokinesis. The thermosensitive *cdc15-140* mutation prevents cytokinesis at the restrictive temperature (*Balasubramanian et al., 1998*). Double mutant cells *cdc15-140 cut14-208* exhibited chromatin bridges during anaphase at 36°C (*Figure 5B*), which indicates that *cdc15-140* did not suppress the chromosome segregation defect caused by the *cut14-208* mutation. However, in the absence of a cytokinetic ring, these chromatin bridges were no longer

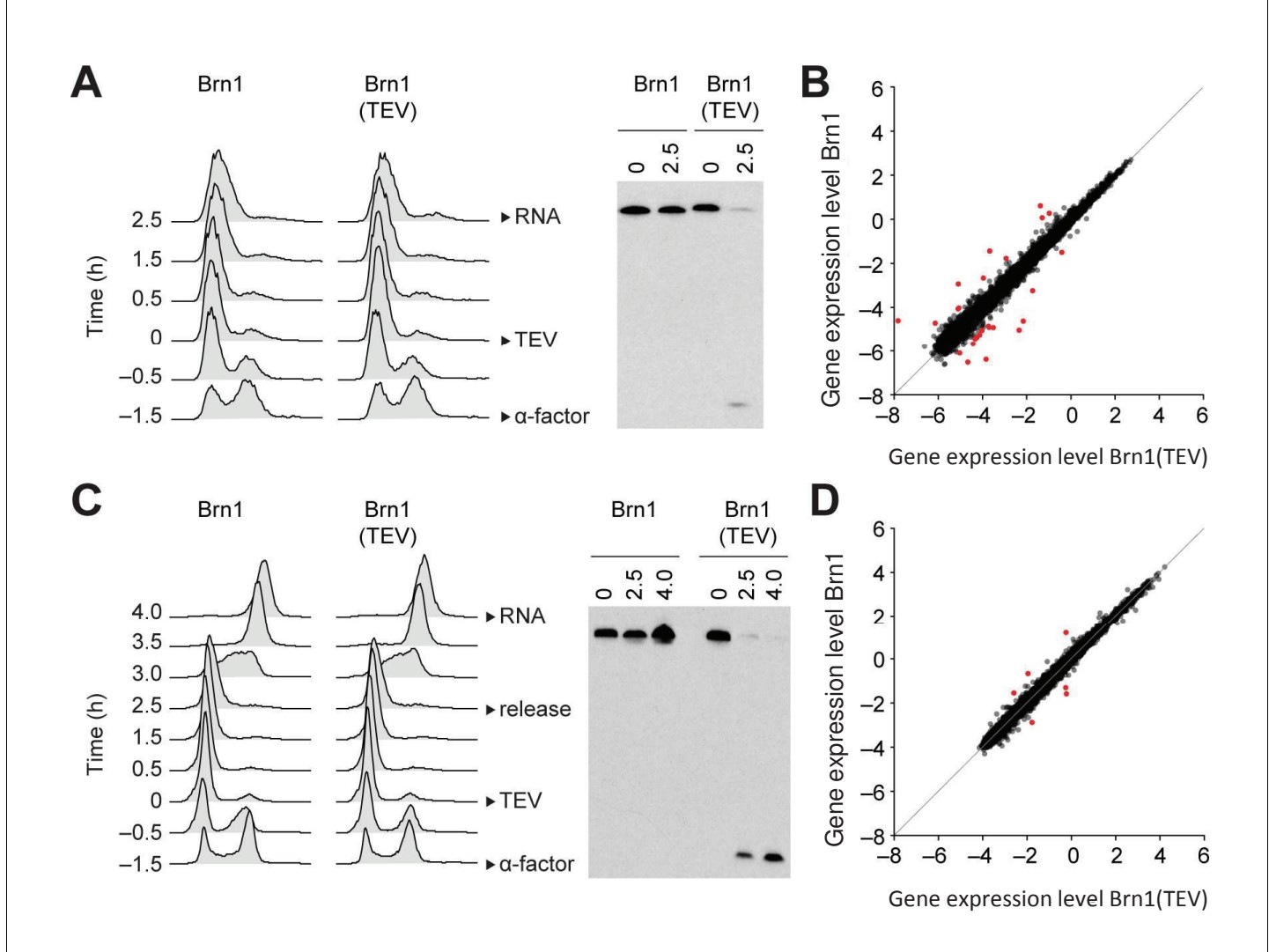

**Figure 4.** Condensin release from chromosomes has no major effects on G1 or M phase gene expression programs in budding yeast. (**A**) TEV protease expression was induced in cells synchronized in G1 phase by α-factor (strains C3138 and C3139). 2.5 hr after TEV induction, RNA was extracted, cDNA synthesized, labelled and hybridized to tiling arrays. Cell cycle synchronization was scored by FACScan analysis of cellular DNA content and Brn1 cleavage was monitored by western blotting against the C-terminal HA$_6$ tag. (**B**) Scatter plot of gene expression values of cells from (**A**) with cleaved or intact Brn1 (mean values of n = 3 biological replicates). Red color highlights two-fold or more up- or downregulated transcripts. (**C**) TEV protease expression was induced in cells synchronized in G1 phase by α-factor (strains C2335 and C2455). Cells were release into nocodazole 2.5 hr after TEV induction and RNA was extracted 1.5 hr later, cDNA synthesized, labelled and hybridized on tiling arrays. Cell cycle synchronization and Brn1 cleavage was monitored as in A. (**D**) Scatter plot of gene expression values of cells from (**C**) with cleaved or intact Brn1 (mean valued of n = 2 biological replicates). Red color highlights two-fold or more up- or downregulated transcripts.

DOI: https://doi.org/10.7554/eLife.38517.008

The following figure supplement is available for figure 4:

**Figure supplement 1.** Condensin release from chromosomes has no major effects on G1 or M phase gene expression programs in budding yeast.
DOI: https://doi.org/10.7554/eLife.38517.009

severed upon mitotic exit. Instead, since fission yeast cells undergo a closed mitosis, chromatin bridges collapsed into a single nucleus as cells exited mitosis. As a consequence, the gain or loss of chromosomal fragments and the production of post-mitotic cells with unbalanced karyotypes were suppressed when cytokinesis was prevented (*Figure 5B*). Remarkably, parallel RNA-seq analysis revealed that ~98% of the RNAs up-regulated in the *cut14-208* single mutant were no-longer

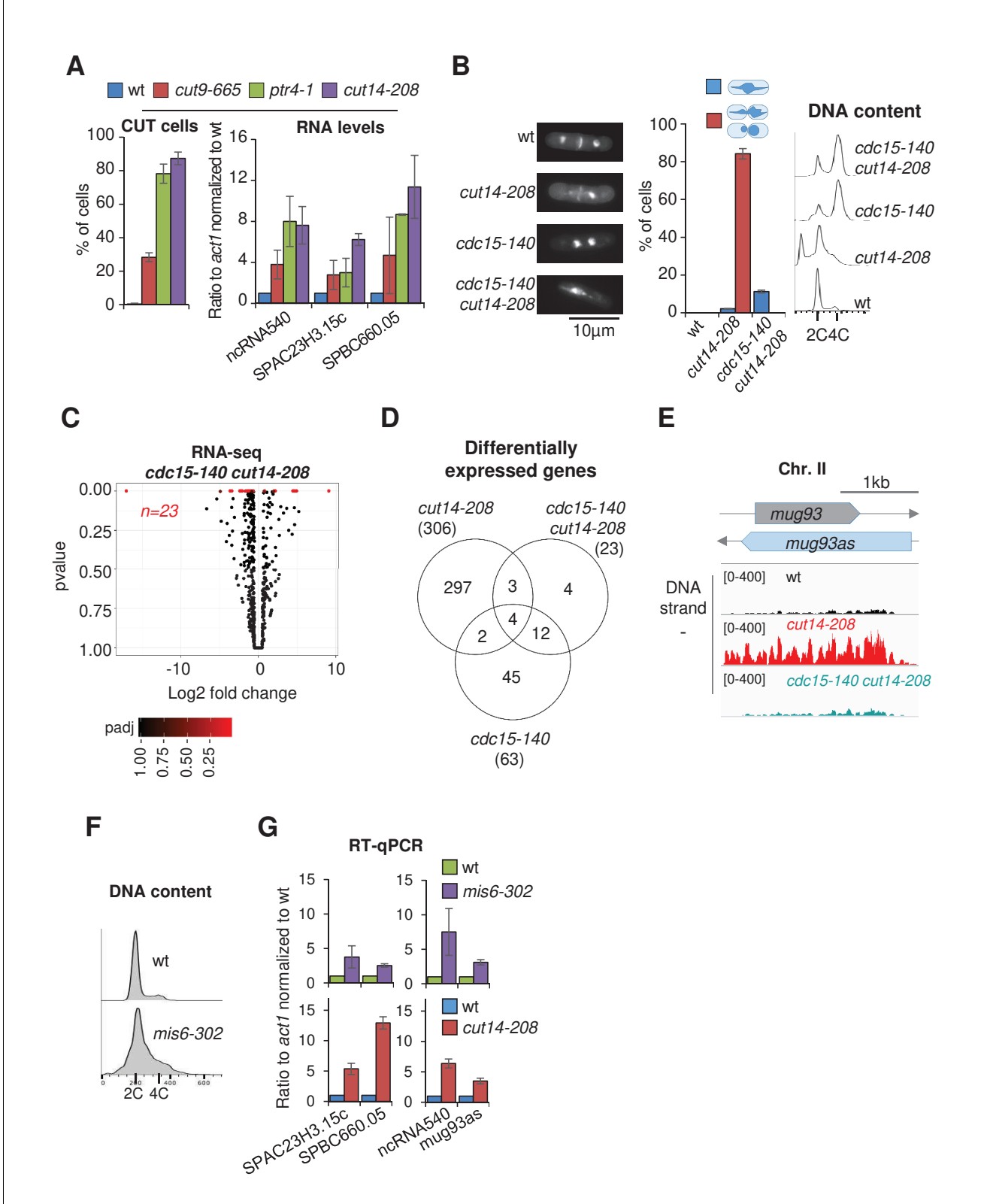

**Figure 5.** Defective mitosis underlies deregulated gene expression in the fission yeast *cut14-208* condensin mutant. (**A**) Gene deregulation in mutant cells in which chromosomes are cut by the cytokinetic ring upon mitotic exit. Strains grown at 36°C for 2.5 hr were processed for cytological analysis and RT-qPCR. Right: cells were stained with DAPI and calcofluor to visualise DNA and the septum, respectively, and to quantify the frequency of chromosome cutting by the septum (CUT cells). Left: total RNA was reverse-transcribed in the presence or absence RT and cDNA quantified by qPCR.
*Figure 5 continued on next page*

*Figure 5 continued*

Shown are averages ± SDs calculated from three biological replicates. (**B–E**) Preventing chromosome severing restores normal gene expression in the condensin mutant *cut14-208*. (**B**) Cells were grown at 36°C for 2.5 hr and stained with DAPI and calcofluor to reveal DNA and the septum, and measure the frequency of CUT cells, or treated for FACScan analysis of DNA content. (**C**) Volcano plot of RNA levels measured by strand-specific RNA-seq in the *cdc15-140 cut14-208* double mutant after 2.5 hr at 36°C, from biological triplicates. (**D**) Comparative RNA-seq transcriptomic analysis from biological triplicates. (**E**) RNA-seq profiles of the *mug93as* ncRNA. (**F–G**) The kinetochore mutation *mis6-302* and the condensin mutation *cut14-208* deregulate a same set of genes. Wildtype and *mis6-302* cells grown at 36°C for 8 hr were processed to analyse DNA content by FACScan (**F**) and RNA levels by RT-qPCR (**G**). *cut14-208* cells and the isogenic wt control grown at 36°C for 2.5 hr were used for comparison. Shown are averages ± SDs measured from biological triplicates.

DOI: https://doi.org/10.7554/eLife.38517.010

The following figure supplements are available for figure 5:

**Figure supplement 1.** Defective mitosis underlies deregulated gene expression in the fission yeast *cut14-208* condensin mutant.

DOI: https://doi.org/10.7554/eLife.38517.011

**Figure supplement 2.** Defective mitosis underlies deregulated gene expression in the fission yeast *cut14-208* condensin mutant.

DOI: https://doi.org/10.7554/eLife.38517.012

detected as differentially expressed in the *cut14-208 cdc15-140* double mutant (**Figure 5C,D**). The suppressive effect of *cdc15-140* on *cut14-208* with respect to the accumulation of the anti-sense RNA *mug93as* is shown as an example in **Figure 5E**. The production of read-through transcripts was similarly suppressed (**Figure 5—figure supplement 2A,B**). Note that RNA levels remained increased in the *cdc15-140 rrp6Δ* mutant (**Figure 5—figure supplement 2C**), ruling out a potential compensatory effect of *cdc15-140* on Rrp6 deficiency per se. Finally, we found that *cdc12-112*, another mutation that also impairs cytokinesis (**Chang et al., 1997**), equally restored normal gene expression in the *cut14-208* genetic background (**Figure 5—figure supplement 2D**), which confirms that cytokinesis was a driving force for the gene deregulation exhibited by the *cut14-208* mutant. Taken together, these data indicate that changes in mRNA and ncRNA levels exhibited by *cut14-208* condensin mutant cells are mostly, if not entirely, the consequence of chromosome cleavage during cytokinesis.

The severing of chromosomes by the cytokinetic ring leads not only to DNA damage, as revealed by the accumulation of sustained Rad22-GFP foci (**Figure 7—figure supplement 1A**), but also to the formation of genomically unbalanced daughter cells, as shown by FACScan analysis of DNA contents (**Figure 3B**). Both phenotypes coincide with the increase in RNA levels, and both are suppressed by *cdc15-140* (**Figure 5B**, **Figure 7—figure supplement 1A**). To test whether one or both might be responsible for gene deregulation in *cut14-208* cells, we analysed by RT-qPCR the impact of DNA damage upon mitotic exit, or of genomic imbalance, on gene expression in cells provided with a fully functional condensin. To damage DNA, we synchronized cells in prometaphase by using the cryosensitive mutation *nda3-KM311* and then released them into mitosis in the presence of the drug Camptothecin to induce DNA breaks during S phase, which, in fission yeast, overlaps with cytokinesis and septum formation. In an alternative experiment, we treated wild-type cycling cells with Zeocin. The appearance of sustained Rad22-GFP foci in cells treated with Camptothecin or Zeocin confirmed the accumulation of DNA damage (**Figure 7—figure supplement 1B**). However, RT-qPCR revealed no increase in ncRNA or mRNA levels (**Figure 7—figure supplement 1B**), arguing that DNA damage is not the main driver for gene deregulation in *cut14-208* mutant cells. To test the impact of genomic imbalance on gene expression, we used the thermosensitive *mis6-302* mutation which disrupts kinetochore assembly, and hence accurate chromosome segregation during anaphase, leading to the production of aneuploid post-mitotic cells (**Saitoh et al., 1997**). Note that *mis6-302* causes neither chromatin bridges nor a CUT phenotype. FACScan analysis of *mis6-302* cells grown at the restrictive temperature confirmed the appearance of genomically unbalanced cells (**Figure 5F**). Remarkably, RT-qPCR of four reporter RNAs revealed a similar increase in *mis6-302* cells as in *cut14-208* mutant cells (**Figure 5G**). This argues that chromosome missegregation during mitosis, which results from condensin inactivation, can lead to gene deregulation.

It has been reported that the expression of six different families of tRNAs, as measured by RT-PCR, is modestly increased in *cut3-477* condensin mutant cells grown at the restrictive temperature of 36°C (**Iwasaki et al., 2010**). We sought to revisit this observation and examined our RNA-seq data sets for tRNA levels. However, we obtained too few exploitable reads mapping within repeated

tRNA genes, possibly because small, structured and extensively post-transcriptionally-modified tRNAs necessitate dedicated protocols to be analysed by RNA-seq (*Wilusz, 2015*). We therefore assessed tRNA levels by RT-qPCR. We selected five reporter tRNAs, amongst which three – SER, SER+MET and LEU – have previously been found to be up-regulated in *cut3-477* condensin mutant cells (*Iwasaki et al., 2010*). We measured the steady state levels of these five families of tRNAs in six different condensin mutant strains – *cnd1-175*, *cut3-i23*, *cut3-477*, *cut3-m26*, *cut14-90* and *cut14-208* – ranked by increasing frequency of chromosome missegregation (*Figure 5—figure supplement 1D*). Although we could not detect any significant change in tRNA levels in *cut3-477* cells, we observed a clear increase in *cut3-m26*, *cut14-90* and *cut14-208* cells (*Figure 6A*). The increase correlated with the frequency of chromosome missegregation and with the accumulation of mRNAs and ncRNAs (compare with *Figure 5—figure supplement 1D,E*). The *cut14-208* mutation exhibited the strongest penetrance. Importantly, preventing aneuploidy in *cut14-208* mutant cells by using the *cdc15-140* mutation restored a normal steady state level of these tRNAs (*Figure 6B*). Conversely, non-condensin mutations that disrupted chromosome segregation, such as *ptr4-1*, *cut9-665* or *mis6-302*, resulted in a similar increase in tRNA levels as in *cut14-208* (*Figure 6C,D*). We noticed, however, that the effects of *ptr4-1* and *cut9-665* were more comparable to the *cut14-208* mutation than to the *mis6-302* mutation, which did not upregulate the tRNAs SER +MET and THR (*Figure 6D*). This difference between *cut14-208* and *mis6-302* is unlikely to be caused by DNA damage that occurs in *cut14-208* cells upon mitotic exit, as neither Camptothecin nor Zeocin treatment increased tRNA levels (*Figure 7—figure supplement 1B*). It might rather be the manifestation of an inherent phenotypic variability associated with the aneuploid condition (*Beach et al., 2017*) (see Discussion). Nevertheless, taken together, these results strongly suggest that the imbalance in genomic content that results from chromosome missegregation during mitosis is a major cause of the deregulation of most, if not all, cellular RNAs when the function of condensin is impaired.

## Non-disjunction of the rDNA during anaphase depletes the RNA-exosome from post-mitotic fission yeast cells and results in ncRNA accumulation

To investigate further the mechanism of gene deregulation when condensin is impaired, we characterized at higher detail chromosome segregation in *cut14-208* mutant cells. It had been reported that condensin plays a major role in the segregation of the rDNA during late anaphase in budding yeast (*Freeman et al., 2000*; *Sullivan et al., 2004*). We reached the same conclusion in fission yeast, confirming previous observations (*Nakazawa et al., 2008*). When we scored the segregation of a GFP-labelled version of the rDNA-binding protein Gar2, we found that sister rDNA copies failed to separate during anaphase in *cut14-208* mutant cells, frequently resulting in the formation of anucleolate daughter cells (*Figure 7A*). The *ptr4-1* and *mis6-302* mutants also exhibited a high rate of anucleolate cell formation (*Figure 7A*), whereas the *cdc15-140* mutation suppressed the formation of anucleolate cells in the *cut14-208* mutant (*Figure 7—figure supplement 1C*). Thus, the inability to properly segregate the rDNA during anaphase correlates with the accumulation of mRNAs and ncRNAs targeted by the RNA-exosome.

Live imaging of Rrp6 and Dis3 had shown that the two RNases are enriched in the nucleolus of fission yeast cells (*Yamanaka et al., 2010*). Given the overlap between the differential transcriptomes of *cut14-208* and *rrp6Δ* mutants, we hypothesised that rDNA non-disjunction might alter the localisation of Rrp6 in daughter cells upon mitotic exit. Co-immunostaining of Rrp6 tagged with a myc epitope and Gar2 tagged with GFP confirmed the nuclear localisation of Rrp6 and its marked enrichment within the nucleolus (*Figure 7B*). In a wild-type background, Rrp6 appeared evenly distributed between nuclei in post-mitotic daughter cells (median signal ratio ~1). In sharp contrast, the amount of Rrp6 was markedly reduced in *cut14-208* anucleolate daughter cells compared to their nucleolate counterparts (median signal ratio ~0.25; *Figure 7B*). We observed a similar asymmetric distribution of Dis3 tagged with an HA epitope in post-mitotic *cut14-208* cells (*Figure 7—figure supplement 1D*). Residual levels of Rrp6 and Dis3 proteins remained detectable in anucleolate cells (*Figure 7B*, *Figure 7—figure supplement 1D*), which suggests that a small fraction of Rrp6 and Dis3 molecules is transmitted to daughter cells independently of the rDNA. Live cell imaging in budding yeast furthermore demonstrated that the segregation into the daughter cells of a mNeonGreen-tagged version of Dis3 was severely impaired during the first cell division after condensin inactivation by Brn1 cleavage (*Figure 7C*, *Figure 7—figure supplement 2*). We conclude that condensin

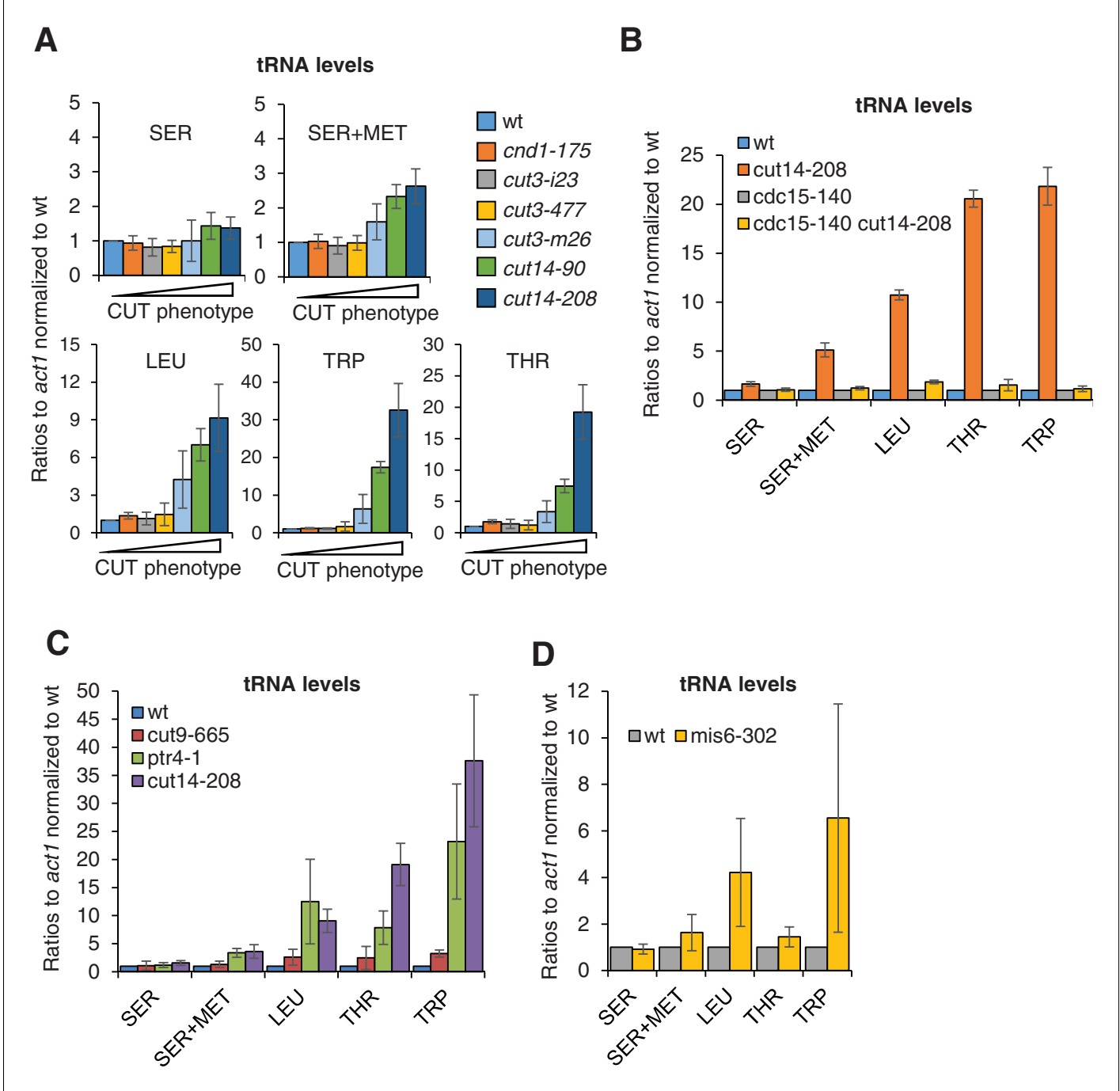

**Figure 6.** The accumulation of tRNAs in condensin mutant cells is linked to chromosome missegregation. (A) The accumulation of tRNAs in condensin mutant cells correlates with chromosome missegregation. Cells were grown at 36°C for 2.5 hr and processed for cytological analysis to measure the frequency of chromosome segregation and CUT phenotype (see *Figure 5—figure supplement 1D*). Total RNA extracted from these cells was reverse-transcribed in the presence or absence of RT and the cDNA were quantified by qPCR with primers unique to the indicated families of tRNAs. SER +MET refers to a dimeric tRNA(SER)-tRNA(MET) transcript (*Johnson et al., 1989*). (B–C) Total RNA extracted from cells grown at 36°C for 2.5 hr was processed for RT qPCR. (D) Wildtype and *mis6-302* cells grown at 36°C for 8 hr were processed for RT-qPCR. *cut14-208* cells and the isogenic wt control grown at 36°C for 2.5 hr were used for comparison. All the data shown in *Figure 6* are averages ± SDs measured from biological triplicates.
DOI: https://doi.org/10.7554/eLife.38517.013

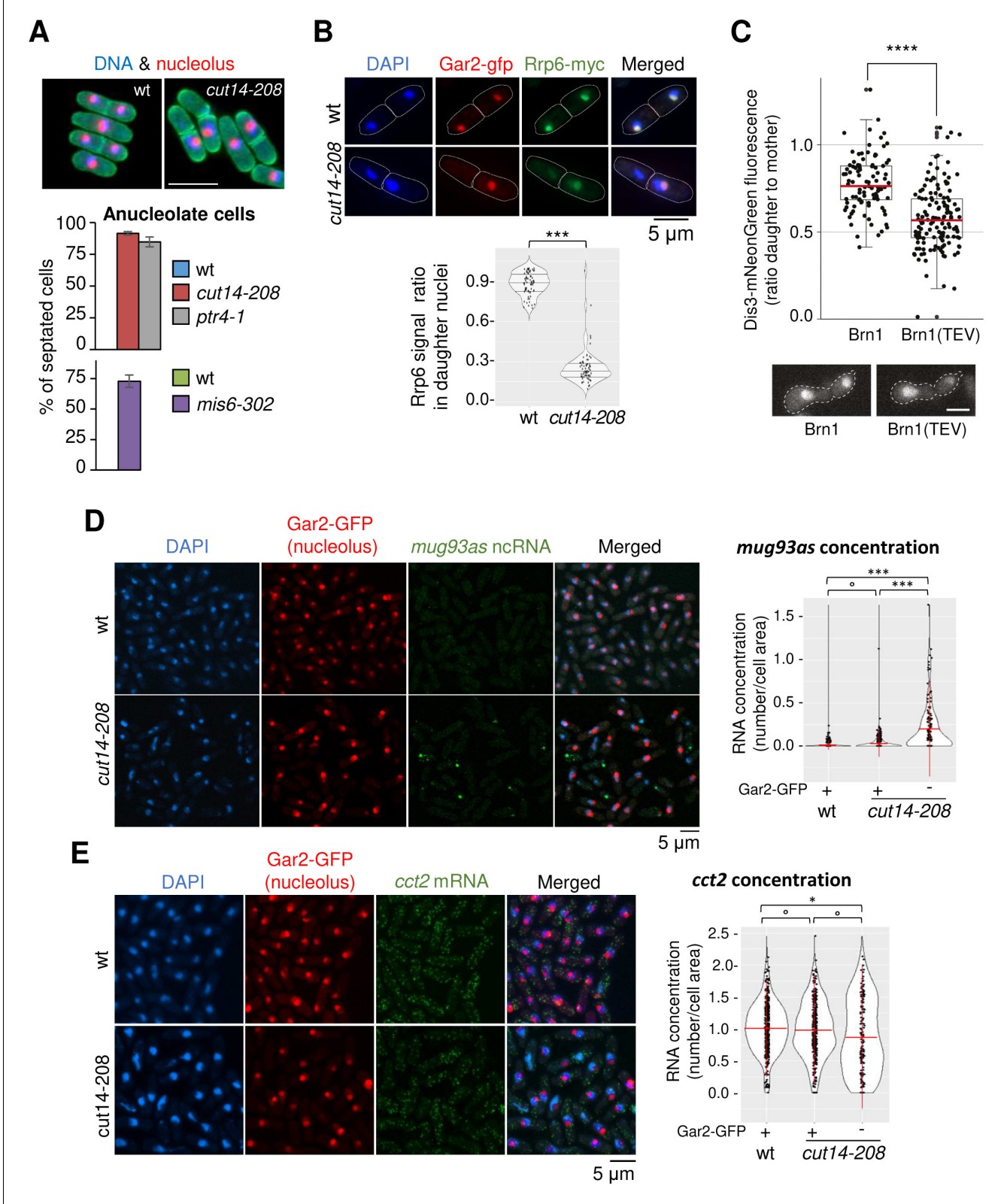

**Figure 7.** Condensin inactivation generates anucleolate daughter cells, which are depleted of the RNases Rrp6 and Dis3 and accumulate unstable RNA. (A) Non-disjunction of the rDNA in *cut14-208* cells. The nucleolar protein Gar2-mcherry was used as a marker for the rDNA (nucleolus) and the plasma membrane protein Psy1-GFP to visualise cytokinesis. Mutant cells and their isogenic wt controls were grown at 36°C for 2.5 hr (*cut14-208* and *ptr4-1*) or 8 hr (*mis6-302*), fixed and stained with DAPI. Segregation of the rDNA in daughter nuclei was measured upon mitotic exit. Scale bar: 10 μm. (B) Rrp6 is

*Figure 7 continued on next page*

*Figure 7 continued*

enriched in the nucleolus, and depleted from anucleolate *cut14-208* mutant cells. Indicated cells were grown at 36°C, fixed and processed for immunofluorescence against Gar2-GFP and Rrp6-myc. DNA was stained with DAPI. Lower panel shows the ratio of Rrp6-myc signals measured within daughter nuclei in septated cells. ***p<0.001, Wilcoxon ranked sum test with continuity correction. (C) Asymmetric partitioning of Dis3 in budding yeast upon condensin cleavage. pTEV protease expression was induced in cells synchronized in G1 phase by α-factor. 2.5 hr after TEV induction, cells were released from the arrest and segregation of Dis3-mNeonGreen was scored between 2 and 2.5 hr after the release by measuring the ratio between Dis3 signals in daughter to mother cells (N = 97 (strain C5259), N = 139 (strain C5260); box plot shows median (red) ±first quartile and 1.5x interquartile range (whiskers); ****p<0.0001, unpaired t-test with Welch's correction). Scale bar: 5 μm. (D–E) The non-coding RNA *mug93as* accumulate in anucleolate *cut14-208* cells. Cells of indicated genotype and expressing Gar2-GFP were grown at 36°C for 2.5 hr, fixed and processed for single molecule RNA FISH using probes complementary to the ncRNA *mug93as* (D) or the mRNA *cct2* (E). Box and whiskers plots show quantifications of RNA spots in *cut14-208* compared to wt, and in nucleolate compared to anucleolate mutant cells. ***p<0.001, *p<0.05 and °p>0.05, Wilcoxon non-parametric test.

DOI: https://doi.org/10.7554/eLife.38517.014

The following figure supplements are available for figure 7:

**Figure supplement 1.** Condensin inactivation generates anucleolate daughter cells, which are depleted of the RNases Rrp6 and Dis3 and accumulate unstable RNA

DOI: https://doi.org/10.7554/eLife.38517.015

**Figure supplement 2.** Condensin inactivation generates anucleolate daughter cells, which are depleted of the RNases Rrp6 and Dis3 and accumulate unstable RNA.

DOI: https://doi.org/10.7554/eLife.38517.016

deficiency leads to rDNA non-disjunction during anaphase and the production of anucleolate post-mitotic cells, which are depleted of Rrp6 and Dis3.

The depletion of Rrp6 and Dis3 provides a plausible cause for the accumulation of exosome-sensitive RNAs in asynchronously dividing *cut14-208* condensin mutant cells. If this were indeed the case, exosome-sensitive RNAs should accumulate preferentially in anucleolate *cut14-208* cells. Single molecule RNA-FISH showed that the ncRNA *mug93as* produced very faint signals in wild-type cells (*Figure 7D*), consistent with its active degradation by the RNA-exosome (*Figure 1D*). On the contrary, *mug93as* levels were considerably higher in *cut14-208* mutant cells (*Figure 7D*), which confirms our previous RNA-seq and RT-qPCR data. Crucially, *mug93as* RNA accumulated mainly in anucleolate cells devoid of Gar2-GFP, with the highest signals in anucleolate cells with low chromatin amount, as judged by DAPI staining. In sharp contrast, a control mRNA (*cct2*) was evenly distributed between the daughter cells in the *cut14-208* mutant (*Figure 7E*). We conclude that condensin loss-of-function leads to the formation of anucleolate cells that accumulate RNA-exosome-sensitive transcripts.

## Discussion

Condensin I and II have been implicated in the control of gene expression in a wide range of organisms, but it has remained unclear what aspect of the gene expression program they affect. Here, we challenge the idea that condensin complexes directly regulate transcription by providing compelling evidence that the functional integrity of condensin is dispensable for the maintenance of proper gene expression during interphase and even mitosis in fission and budding yeasts. Consistent with previous studies, we show that condensin deficiency alters the transcriptome of post-mitotic cells in fission yeast. However, this effect is mostly, if not entirely, the indirect consequence of chromosome missegregation during anaphase. Our findings therefore indicate that condensin plays no direct role in the control of gene expression in fission and budding yeasts, but is essential for the maintenance of proper gene expression by contributing to the accurate segregation of chromosomes during mitosis.

Strand-specific RNA-seq and RT-qPCR analyses reveal that *cut14-208* condensin mutant cells accumulate tRNAs, mRNAs, ncRNAs and 3'-extended read-through transcripts. Other condensin mutants such as *cut3-m26*, *cut14-90* and *cut14-180* exhibit similar increases in RNA levels, arguing that condensin takes part in proper gene expression in fission yeast, similar to other organisms. The population of *cut14-208* condensin mutant cells that exhibited increased RNA levels went through G2, M, G1, and S phases at the restrictive temperature in an asynchronous manner before their RNA was extracted and analysed. However, when *cut14-208* cells were synchronised at S or G2 phase or

even at metaphase, RNA levels remained unchanged. Moreover, the prevalent suppressive effect of *cdc15-140* upon *cut14-208* on changes in gene expression argues that condensin impinges on gene expression in a cytokinesis-dependent manner, and therefore in an indirect manner. Importantly, it also implies that condensin deficiency makes by itself no predominant impact on the steady state level of RNAs in post-mitotic cells. Thus, although we cannot rule out the possibility that subtle impacts of condensin on the dynamics of transcription might have escaped our detection, our data strongly indicate that condensin plays no major direct role in the maintenance of proper gene expression during interphase or during mitosis in fission yeast. Concordantly, we show that also in budding yeast, condensin is largely dispensable for the maintenance of gene expression programs during the G1 phase or during metaphase. Importantly, our results are in agreement with a recent report that the global transcriptional program of *S. cerevisiae* is largely insensitive to condensin depletion (*Paul et al., 2018*). Note that the separation of the budding and fission yeast lineages is thought to have occurred ~420 million years ago, which makes them as distant from each other as either is from animals (*Sipiczki, 2000*). Thus, in two evolutionarily distant organisms, condensin complexes play no major direct role in the maintenance of an established gene expression program.

The corollary is that the increased RNA levels exhibited by fission yeast cells defective for condensin must be the indirect consequence of a failure during late mitosis caused by a lack of condensin activity. Indeed, RNA levels increase coincidently with mitotic exit in synchronized condensin mutants and the amplitude of the increase correlates with the prevalence of chromosome missegregation during anaphase. Furthermore, suppressing the production of genetically unbalanced daughter cells by the cytokinesis mutation *cdc15-140* is sufficient to restore an almost normal transcriptome, despite condensin being impaired by the *cut14-208* mutation. Reciprocally, preventing proper chromosome segregation by mutations in separase (*ptr4-1*), the APC/C (*cut9-665*) or the kinetochore (*mis6-302*) is sufficient to trigger the accumulation of aberrant transcripts alike in *cut14-208* cells. Therefore, the missegregation of chromosomes during anaphase appears to be the principal responsible mechanism for the changes in RNA levels exhibited by condensin mutant cells.

A clear illustration of the impact of chromosome missegregation is provided by the non-disjunction of the duplicated copies of the rDNA, which depletes the RNA-exosome from daughter cells and leads to the accumulation of cellular RNAs. Indeed, we show (1) that sister-rDNAs systematically fail to disjoin during anaphase in *cut14-208* condensin mutant cells, leading to the production of anucleolate daughter cells; (2) Rrp6 and Dis3, which are enriched in the nucleolus, become depleted from anucleolate *cut14-208* cells; (3) the reduction of Rrp6 and Dis3 from anucleolate condensin mutant cells coincides with an increased steady state level of RNA-exosome-sensitive transcripts; and (4) an RNA-exosome sensitive transcript, the antisense RNA *mug93as,* accumulates principally if not exclusively in anucleolate condensin mutant cells, as expected from an impaired degradation. It is important to note that Rrp6 and Dis3 are reduced but not eliminated from anucleolated *cut14-208* condensin mutant cells. Knowing that Rrp6 accumulates predominantly on chromosomes during mitosis in *Drosophila* (*Graham et al., 2009*), we propose that the bulk of Rrp6 and Dis3 might similarly co-segregate with the nucleolus in fission yeast, while a fraction is presumably transmitted to daughter cells in association with chromatin or within the nucleoplasm. The co-depletion of Rrp6 and Dis3 from anucleolate *cut14-208* cells provides a straightforward explanation as to why the transcriptomes of the *cut14-208* and *rrp6Δ* strains overlap only partially, and to the slight additive effect exhibited by the *cut14-208* and *rrp6Δ* mutations with respect to the accumulation of *mug93as* (*Figure 1D*) and of 3'-extented *hsp9* RNA (*Figure 2B*). It can also explain why *mug93as* RNA accumulated to the highest levels in anucleolate cells with low chromatin amount (*Figure 7D*).

Based on these observations, we propose the model depicted in *Figure 8*. Condensin deficiency impairs chromosome segregation during anaphase, and notably leads to chromatin bridges and the non-disjunction of the rDNA. Upon mitotic exit, this failure to properly segregate chromosomes alters the karyotype of daughter cells and creates an aneuploid condition that changes the cellular RNA content. In particular, Rrp6 and Dis3 which are enriched in the nucleolus during mitosis become depleted from anucleolate daughter cells. This reduces the activity of the RNA-exosome, allowing the accumulation of RNA molecules that are normally actively degraded by the RNA-exosome, such as ncRNAs and 3'-extended RNAs. This sequence of events illustrate how chromosome missegregation caused by condensin deficiency indirectly changes cellular RNA contents in fission yeast.

Aneuploid cells exhibit gene-specific deregulations along a more stereotypical 'aneuploid stress-response' characterised by an increased expression of genes involved in the response to stress and

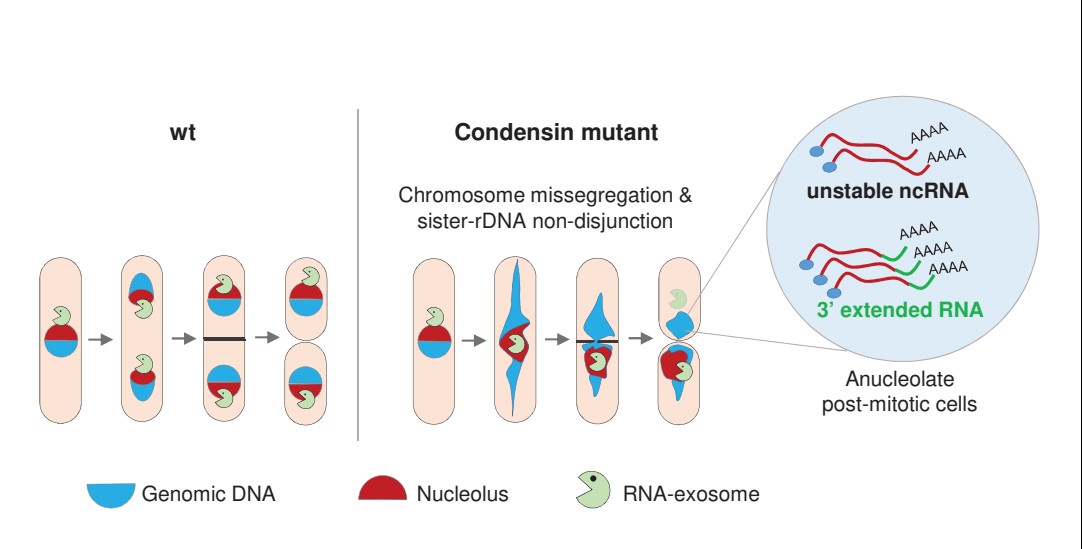

**Figure 8.** Condensin deficiency impinges upon gene expression by promoting accurate chromosome segregation throughout mitosis. In wild-type fission yeast cells, Rrp6 and Dis3, the catalytic subunits of the RNA-exosome, are enriched in the nucleolus and a large fraction co-segregates with the bulk of the rDNA during anaphase. In condensin mutant cells, chromosomes fail to properly segregate during anaphase. Chromatin bridges are formed and entangled sister-rDNA copies fail to separate, leading to the production of karyotypically aberrant cells, whose aneuploid condition changes the transcriptome. Notably, the non-disjunction of the rDNA co-depletes anucleolate daughter cells from Rrp6 and Dis3, allowing the accumulation of RNA molecules that are normally actively degraded by the RNA-exosome, such as unstable ncRNAs and 3'-extended RNAs.
DOI: https://doi.org/10.7554/eLife.38517.017

a down-regulation of cell-cycle and proliferation genes (*Santaguida and Amon, 2015*; *Sheltzer et al., 2012*). Premalignant murine T-cells defective for condensin II exhibit a transcriptome evocative of a stress response to aneuploidy (*Woodward et al., 2016*). Similarly, the depletion of Smc2, and hence the reduction of both condensin I and II, in a human neuroblastoma cell line modifies the expression of a large number of genes implicated mainly in cell cycle progression or DNA damage response (*Murakami-Tonami et al., 2014*). However, we did not observed such a clear transcriptomic signature in *cut14-208* mutant cells. One possible explanation is that condensin deficiency causes mostly the severing of chromosome arms by the cytokinetic ring in fission yeast, which presumably leads to the gain or loss of chromosomal fragments rather than of entire chromosomes. The resulting partial- or micro-aneuploidy (*Pfau and Amon, 2012*) might, in contrast to a whole chromosomal aneuploidy, not elicit a robust aneuploidy stress response, which presumably stems from the concomitant change in number of many genes that have little impact when varied individually (*Santaguida and Amon, 2015*). That different types of aneuploidy might have varying impacts on the transcriptome might also explain the slight differences we observed between the tRNA contents of *cut14-208* and *mis6-302* mutant cells, which missegregate entire chromosomes. In any cases, multiple mechanisms linked to chromosome instability and the genesis of aberrant karyotypes most likely concur to modify the transcriptome when condensin is defective. For instance, the slight up-regulation of histone genes that we observed in *cut14-208* mutant cells has previously been attributed to an indirect stabilization of the transcription factor Ams2 during defective anaphase (*Kim et al., 2016*).

Our finding that condensin prevents the accumulation of 3' extended read-though RNAs is reminiscent of the role played by cohesin in the termination of transcription at convergent genes. Probably pushed by active RNA Pol II, cohesin accumulates at intergenic regions that are flanked by co-transcribed convergent genes, where it prevents transcription to proceed beyond canonical termination sites (*Gullerova and Proudfoot, 2008*). It is important to note that in the case of unidirectional transcription, read-through RNAs are produced in a cohesin–independent manner (*Gullerova and Proudfoot, 2008*), implying that cohesin plays the role of a roadblock for RNA polymerases specifically at the 3'ends of co-transcribed convergent genes. Our observation that cohesin remains bound to chromatin in *cut14-208* cells, including at convergent genes that exhibit read-through transcripts,

plus the lack of any noticeable increase in Pol II occupancy, argue that condensin is unlikely to prevent read-thought transcripts through a same mechanism. The fact that 43% of read-through RNAs originate from genes oriented in tandem in *cut14-208* cells further support this conclusion. Thus, besides cohesin-dependent termination, another pathway dependent upon condensin must also exist that prevents read-through transcripts. However, and crucially, condensin is unlikely to play a direct role in this second pathway, since *cdc15-140 cut14-208* double mutant cells no longer accumulate read-through RNAs. Furthermore, given that *cut14-208* reduces the binding of condensin to chromatin (*Nakazawa et al., 2015*), the lack of read-through RNAs in *cdc15-140 cut14-208* cells suggests that condensin is unlikely to exert any major roadblock effect on active RNA polymerases. The overlap between *cut14-208* and *rrp6Δ* cells suggests instead that it is the degradation of read-through transcripts that is impaired. Read-through transcripts are produced in wild-type cells (*Gullerova and Proudfoot, 2008*; *Zofall et al., 2009*). We therefore propose that read-through RNAs are naturally produced both at tandem and convergent genes in wild-type cells, but are rapidly degraded by Rrp6. Chromosome missegregation during mitosis caused by condensin deficiency induces their accumulation, despite the presence of cohesin, at least in part by depleting Rrp6 from karyotypically abnormal daughter cells.

Our finding that sister-rDNA almost systematically non-disjoin during anaphase in fission yeast condensin mutant cells is consistent with the severe missegregation of the rDNA observed in budding yeast and human cells deprived of functional condensin complexes (*D'Ambrosio et al., 2008b*; *Freeman et al., 2000*; *Samoshkin et al., 2012*), and with the idea that decatenation of sister-rDNA during late anaphase relies upon condensin (*D'Ambrosio et al., 2008a* ). In that context, the asymmetric partitioning of the ribonuclease Dis3 that we observed in budding yeast daughter cells after the inactivation of condensin by TEV cleavage can be reasonably interpreted as a consequence of a failure to disjoin sister-rDNA during anaphase. In addition, it has been reported that mutations affecting condensin generate anucleolate cells that accumulate polyadenylated RNAs in budding yeast (*Paul and Montpetit, 2016*). Therefore, the missegregation of the rDNA that frequently occurs during anaphase when condensin is impaired presumably impinges upon gene expression in budding yeast, alike in fission yeast.

One related aspect that remains puzzling, however, is that chromatin bridges are almost systematically severed by the cytokinetic ring in *cut14-208* mutant cells, while the severing of the nucleolus is extremely infrequent (<10% of the cases). The persistence of the nucleolus in the axis of cleavage might mechanically hinder cytokinesis, or trigger a wait signal. Alternatively, condensin deficiency might cause DNA double-strand breaks at fragile sites located on the centromere-proximal border of the cluster of rDNA repeats, allowing the displacement of untangled sister-rDNA towards one pole of the mitotic spindle through a spring-relaxation effect. A focused role for condensin in organising a region proximal to the rDNA, and located on its centromeric side, has been reported in budding yeast (*Schalbetter et al., 2017*). In HeLa cells, the depletion of SMC2 induces DNA breaks predominantly in repetitive DNA, including the rDNA/Nucleolar Organising Regions (*Samoshkin et al., 2012*). Thus, missegregation of the rDNA in budding and fission yeast cells when condensin is impaired might reveal an evolutionarily-conserved acute dependency of repeated DNA elements upon condensin for their segregation and integrity. Given the prevalence of tandem repetitive DNA arrays in mammalian genomes (*Warburton et al., 2008*), it is tempting to speculate that condensin loss of function might have similar confounding impacts on the preservation of their structural integrity and proper expression.

Studies performed over the past 20 years have shown that the three-dimensional organisation of the genome influences gene expression, raising the key question of the role played by SMC complexes in the control of gene expression. A large number of studies have concluded in favour of a role for cohesin and condensin I and II in the control of gene expression (*Dowen and Young, 2014*; *Merkenschlager and Nora, 2016*), raising the idea that cohesin and condensins might collectively link gene expression to genome architecture. Although there are robust examples of cohesin-mediated regulation of cell-type-specific gene expression (*Merkenschlager and Nora, 2016*), for instance through enhancer-to-promoter interactions (*Ing-Simmons et al., 2015*; *Kagey et al., 2010*), a recent study has raised the idea that the involvement of cohesin in the maintenance of an established gene expression program might be less prominent than initially thought (*Rao et al., 2017*). Similarly, despite ample reports of cells defective for condensin I or II exhibiting changes in gene expression (*Bhalla et al., 2002*; *Dowen et al., 2013*; *He et al., 2016*; *Iwasaki et al., 2010*; *Kranz et al., 2013*;

*Li et al., 2015*; *Longworth et al., 2012*; *Lupo et al., 2001*; *Murakami-Tonami et al., 2014*; *Rawlings et al., 2011*; *Wang et al., 2016*; *Yuen et al., 2017*), to the best of our knowledge, there has thus far been no clear case where the influence of condensin I or II on gene expression has been dissociated from a possible confounding effect of chromosome missegregation. By providing evidence that condensin plays no major direct role in the control of gene expression in fission and budding yeasts, and showing that condensin impinges on gene expression by preserving the stability of the genome during mitosis, our work challenges the concept of gene regulation as a collective property of SMC complexes, and should help to better define future studies on the role of canonical condensins in gene expression in other organisms.

# Materials and methods

**Key resources table**

| Reagent type (species) or resource | Designation | Source or reference | Identifiers | Additional information |
|---|---|---|---|---|
| Genetic background (S. pombe) | 972 h- | NA | 245818 [UID] 391418 [GenBank] 245818 [RefSeq] | |
| Genetic background (S. cerevisiae) | W303 | NA | | |
| Genetic reagents (yeast strains) | See *Supplementary file 2* | | | |
| Antibody | Anti-tubulin (mouse monoclonal) | Keith Gull | TAT1 | (1:50) |
| Antibody | Anti-myc (mouse monoclonal) | Thermo Fisher | #9E10 | (1:500) |
| Antibody | Anti-HA (mouse monoclonal) | Sigma-Aldrich | #12CA5 | (1:500) |
| Antibody | Anti-GFP (rabbit polyclonal) | Life Technologies | #A11122 | (1:800 for IF and 6 μg / IP for ChIP) |
| Antibody | Anti-ser2P RNAPII (rabbit polyclonal) | Abcam | #ab5095 | 6 μg / IP |
| Antibody | Anti-HA (mouse monoclonal) | Covance | 16B12 (anti-HA.11) | 1.5 μl / 50 μl beads |
| Antibody | Anti-PK (mouse monoclonal) | Abd Serotec | MCA1360 | 2 μl / 50 μl beads |
| Sequence-based reagent | See *Supplementary file 3* | Sigma | | |
| Commercial assay or kit | Dynabeads protein A | Invitrogen | #10002D | |
| Commercial assay or kit | In vitro transcription kit for probe labelling | Ambion/Thermo Fisher | T7 riboprobe system (AM1312) | |
| Chemical compound, drug | 3-BrB-PP1 | Toronto Research Canada | A602985 | |
| Software, algorithm | RNA seq analysis | This study | https://github.com/LBMC/readthroughpombe | |

## Media, molecular genetics and strains

Media and molecular genetics methods were as previously described (*Moreno et al., 1991*). Fission yeast cells were grown at 28°C in complete YES + A medium or in synthetic PMG medium. The *nmt1-dis3* chimerical gene was repressed by the addition of thiamine 60 μM final to the growth medium, as described (*Lemay et al., 2014*), followed by further cell culture for 12 – 15 hr. Strains used in this study are listed in *Supplementary file 2*.

## Cell cycle synchronization

Fission yeast cells were synchronized in early S phase at 28°C by the adjunction of hydroxyurea (HU) 15 mM final. G2/M arrest was achieved using the thermo-sensitive *cdc25-22* mutation or the analogue-sensitive *cdc2asM17* allele (*Aoi et al., 2014*), and G1 arrest using the thermo-sensitive *cdc10-129* mutation. Reversible prometaphase arrest was performed at 19°C using the cold-sensitive *nda3-KM311* mutation (*Hiraoka et al., 1984*). Metaphase arrest was achieved in PMG medium supplemented with thiamine 20 μM using the thiamine repressible *nmt41-slp1* gene (*Petrova et al., 2013*). Mitotic indexes were measured as the percentages of mitotic cells accumulating Cnd2-GFP in the nucleus (*Sutani et al., 1999*). Budding yeast cells were grown at 30°C in YEPD to mid-log phase, collected by filtration, washed with ddH$_2$O, and re-suspended at an OD$_{600}$ of 0.30 in YEPD containing 3 μg/ml α-factor. After one hour, additional α-factor was added to 3 μg/ml. After another hour, an aliquot of cells was used for ChIP (G1 sample) and the remaining cells were collected by filtration, washed with dH$_2$O and re-suspended in YEPD to release the cells from the G1 arrest. 30 min after and 60 min after the release, aliquots were collected for ChIP (S phase sample and G2 sample respectively). Budding yeast strains expressing TEV protease under the *GAL1* promoter were grown at 30°C in YEP medium containing 2% raffinose (YEP-R) to mid-log phase, collected by filtration, washed with ddH$_2$O, and re-suspended at an OD$_{600}$ of 0.15 in YEP-R containing 3 μg/ml α-factor. After one hour, additional α-factor was added to 3 μg/ml. After 30 min, the cultures were split and TEV protease expression was induced in one half by addition of 2% galactose. After another 30 min, cells were collected by filtration, washed with dH$_2$O and re-suspended in YEP-R (uninduced) or YEP-R with 2% galactose (YEP-RG; induced) with 3 μg/ml α-factor. Fresh α-factor was added to 3 μg/ml after another hour to all cultures and ChIP samples were collected one hour later.

## Microscopy

To quantify anucleolate cells, cells expressing Gar2-mCherry and Psy1-GFP fusion proteins were fixed with cold methanol and DNA was stained with 4′, 6-diamidino-2-phenylindole (DAPI) at 0.5 μg/ml in PEM buffer (100 mM PIPES, 1 mM EGTA, 1 mM MgSO4, pH 6.9). Gar2-mCherry and Psy1-GFP were directly observed under the fluorescent microscope. Rad22-GFP foci were analysed on cells fixed with iced-cold methanol and stained with DAPI. Cytological analysis of the CUT phenotype was performed as described (*Hagan, 2016*) except that cells were fixed with cold methanol and stained with Hoechst 33342 (20 μg/ml). Immunofluorescence was performed as described (*Robellet et al., 2014*), with the following modifications. Cells shifted at 36°C for 2h30 min were fixed with ice-cold ethanol and stored at 4°C. 2 × 10$^7$ cells were washed in PEMS (PEM +1.2 M Sorbitol) and digested with Zymolyase 100T (0.4 mg/ml in PEMS) for 30 min at 37°C. Images were processed and quantified using ImageJ with automated background subtraction.

Aliquots of budding yeast cells grown in YEP-RG media were transferred to Concavalin A-coated Cellview slides (Advanced TC, Greiner Bio-One). After 20 min, cells were washed twice and covered with SC media containing 2% raffinose and 2% galactose medium for live cell imaging. Images were acquired on an LSM780 with a 63×, 1.4 NA, Oil Plan Apochromat objective (Zeiss) controlled by ZEN 2012 software. The microscope was equipped with an incubation chamber (EMBL) and heated to 30°C. The pinhole was entirely opened and 7 z-slices with a distance of 1 μm were recorded.

Dis3-mNeonGreen fluorescence measurements were performed in Fiji. The z-section with the highest Dis3-mNeonGreen fluorescence was manually determined and a maximum projection of this plus the previous and following slices were performed. All cells where mother and daughter cells were in focus and at least one small bud was present in the mother or daughter cell (to ensure that chromosome segregation had been completed) were marked in the transmission channel and used for analysis. The Dis3 signal was then measured in circular 50 × 50 px ROIs in the mother and daughter cell. After background subtraction (determined as the mean of four 50 × 50 px ROIs in the cell-

free region), the ratio of the Dis3 signal in the daughter cells and mother cell was calculated and plotted. Since samples were normally distributed according to a Kolmogorov-Smirnov test, but variances were different according to Levene's test ($\alpha$ = 0.05), an unpaired t-test with Welch's correction was chosen to test for statistical significance.

## FAScan

$2 \times 10^6$ fission yeast cells were fixed with ethanol 70% (v/v), washed in sodium citrate (50 mM pH 7) and digested with RNase A (100 µg/ml) (Merck). Cells were briefly sonicated and stained with 1 µM Sytox Green (ThermoFischer Scientific). DNA content was quantified on a FACSCALIBUR cytometer using CellQuest Pro software (BD Biosciences). Raw data were analyzed with FlowJo software (BD biosciences). FACScan analysis of budding yeast cells was performed as previously described after staining DNA with either propidium iodide (*Figure 4*; *Cuylen et al., 2011*) or SYBR green I (*Figure 4—figure supplement 1*, *Figure 7—figure supplement 2*; *Cuylen et al., 2013*).

## Chromatin immunoprecipitation and quantitative PCR

ChIP against fission yeast RNA Pol II (S2P) was performed as described (*Vanoosthuyse et al., 2014*) using $2 \times 10^8$ cells fixed with 1% formaldehyde at 36°C for 5 min and then 19°C for 25 min. ChIP against Psm3-GFP was performed on cells fixed with 3% formaldehyde at 36°C for 30 min, as described (*Bhardwaj et al., 2016*). Fixed cells were washed with PBS and lysed using acid-washed glass beads in a Precellys homogenizer. Chromatin was sheared into 300- to 900-bp fragments by sonication using a Diagenode bioruptor. Sheared chromatin was split in two equivalent fractions subjected to parallel immunoprecipitations using magnetic Dynabeads coated with the appropriate antibody. Total and immunoprecipitated DNA was purified using the NucleoSpin PCR clean-up kit (Macherey-Nagel). DNA was analysed on a Rotor-Gene PCR cycler using QuantiFast SYBR Green mix. ChIP-qPCR experiments for budding yeast cells were performed as described previously (*Cuylen et al., 2011*). In brief, aliquots of 42 ml culture with an $OD_{600}$ of 0.6 were fixed in 3% formaldehyde at 16°C. Chromatin was sonicated to an average length of 500 bp using a Diagenode bioruptor. For anti-HA immunoprecipitation, 50 µl protein G dynabeads and 1.5 µl 16B12 antibody (anti-HA.11, Covance) were used. For anti-PK immunoprecipitation, 50 µl protein A dynabeads and 2 µl anti-PK (V5) tag antibody (Abd Serotec MCA1360) were used. Purified DNA was analysed with an ABI 7500 real-time PCR system (Applied Biosystems) using rDNA-specific primers. Primers are listed in *Supplementary file 3*.

## Chromosome spreads

Budding yeast cells were grown at 30°C in YEPD to mid-log phase, collected by filtration, washed with $dH_2O$, and re-suspended at an $OD_{600}$ of 0.2 in YEPD containing 3 µg/ml $\alpha$-factor. After one hour, additional $\alpha$-factor was added to 3 µg/ml. After another hour, an aliquot of cells was used for chromosome spreading (G1 sample) and the remaining cells were collected by filtration, washed with $dH_2O$ and re-suspended into YEPD to release the cells from the G1 arrest. Aliquots were collected for chromosome spreading 30 min and 60 min after the release (S and G2 phase samples, respectively). Chromosome spreads were prepared as described previously (*Cuylen et al., 2011*) and stained for Brn1-HA$_6$ with 16B12 (anti-HA.11, Covance, 1:500) and Alexa Fluor 594–labelled anti-mouse IgG (Invitrogen, 1:600) antibodies and for DNA with DAPI. Images were recorded on a DeltaVision Spectris Restoration microscope (Applied Precision) with a 100×, NA 1.35 oil immersion objective.

## Total RNA extraction and RT-qPCR

Total RNA was extracted from $2 \times 10^8$ fission yeast cells by standard hot-phenol method. 1 µg of total RNA was reverse-transcribed using Superscript III (Life Technologies) following the manufacturer's instructions, using random hexamers in the presence or absence of Reverse Transcriptase (RT). cDNAs were quantified by real time qPCR on a Rotor-Gene PCR cycler using QuantiFast SYBR Green mix. The absence of PCR product in minus RT samples has been verified for all RT-qPCR shown in this publication. Primers are listed in *Supplementary file 3*.

## RNase digestion and northern blot

For RNase H digestion, total RNA was hybridized with a DNA oligonucleotide complementary to a sequence located at the 5' end of the *hsp9* mRNA and digested with RNase H (Roche) following the manufacturer's instructions. For northern blotting, total or RNase H-digested RNA was resolved on 1% agarose gel supplemented with formaldehyde 0.7% (v/v), transferred onto Hybond-XL nylon membranes (Amersham) and cross-linked. Pre-hybridization and overnight hybridization were carried out in ULTRAhyb buffer (Ambion) at 68°C. Strand-specific RNA probes were generated by in vitro transcription using the T7 riboprobe system (Ambion) and internally labelled with [$\alpha$−32P]-UTP. Membranes were quickly washed with 2X SSC and 0.1% SDS, 10 min in 2X SSC and 0.1% SDS, and 3 times in 0.2X SSC and 0.1% SDS. Blots were imaged with Typhoon 8600 instrument (Molecular Dynamics) and quantified with ImageQuant TL (GE Healthcare).

Single molecule RNA Fluorescence In Situ Hybridization (smFISH), imaging and quantification smFISH was performed on formaldehyde-fixed cells, as described (*Keifenheim et al., 2017*). Probes were designed and synthetized by Biosearch Technologies (Petaluma, CA). The *mug93as* and *cct2* probes were labelled with Quasar 670. Probe sequences are listed in *Supplementary file 3*. Cells were imaged on a Leica TCS Sp8, using a 63x/1.40 oil objective, Optical z sections were acquired (z-step size 0.3 microns) for each scan to cover the depth of the cells. For image analysis and quantification, cell boundaries were outlined manually and RNA spots quantified using FISH-quant package implemented in MATLAB, as described (*Mueller et al., 2013*). The FISH-quant detection technical error was estimated at 6 – 7% by quantifying mRNA numbers using two sets of probes covering the 5' half or the 3' half of the *rpb1* mRNA and labelled with different dyes.

## RNA-seq and analysis

RNA-seq was performed on biological triplicates. Total RNA was extracted from $2 \times 10^8$ yeast cells by standard hot-phenol method. RNA quality was determined using TapeStation (Agilent) with RINe score >9. Ribosomal RNA was removed by treating 2 µg of total RNA with the Ribo-Zero Gold rRNA Removal Kit (Yeast) (MRZY1324, Illumina, Paris, France). RNA-seq libraries were prepared using Tru-Seq Stranded kit. Sequencing was performed on Illumina Hiseq 4000, with single-end reads of 50 nt in length. Total number of reads per sample ranged from 59 million to 93 million. For in-silico analyses, we used the version ASM294v2.30 of the *S. pombe* genome in fasta format and the corresponding gff3 annotations downloaded from the ebi website (2017/03/22). Scripts are available in the following git repository: https://github.com/LBMC/readthroughpombe (*Modolo, 2018*; copy archived at https://github.com/elifesciences-publications/readthroughpombe). The RNA-seq reads were processed with cutadapt (*Martin, 2011*) to remove adaptors and further trimmed with UrQt (–t 20) (*Modolo and Lerat, 2015*) based on their quality. After quality control with fastqc (*Andrews, 2010*), we built the reverse complement of the reads in the fastq files using seqkit, indexed the genome (-build) and mapped the trimmed fastq files (–very-sensitive) using Bowtie2 (*Langmead and Salzberg, 2012*). Mapping output files were saved in bam format with samtools (view -Sb) (*1000 Genome Project Data Processing Subgroup et al., 2009*).

To detect read-through events, we searched for sections of reads located immediately downstream of the 3' ends of annotated transcription units, on the same DNA strand, and within gene-free intergenic regions. First, broad peaks of RNA-seq reads were identified using the peak caller Music v6613c53 (*Harmanci et al., 2014*). Bam files were sorted and filtered as forward or reverse with samtools (view -hb -F 0 $\times$ 10 or view -hb -f 0 $\times$ 10). The gff annotation file was converted to bed with convert2bed from bedtools, the genome indexed with samtools (faidx) and the annotation file split into forward and reverse with bedtools (complement). Next, reads corresponding to annotated transcripts were removed using samtools view (-hb bams -L annotation). Subtracted bam files were sorted with samtools sort, and RNA-seq peaks located outside annotated transcripts were detected with Music (-get_per_win_p_vals_vs_FC -begin_l 50 -end_l 500 -step 1.1 l_mapp 50 l_frag 363 -q_val 1 l_p 0). The resulting annotation was further filtered by removing peaks whose starting positions were located more than two read-length away from the nearest 3'end of a transcript. Only peaks detected in at least 2 out of 3 biological replicates were considered. Read-through detection was performed independently for the *rrp6Δ* and *cut14-208* mutants. To achieve comparable examination between *rrp6Δ* and *cut14-208* for reads quantification, we merged their respective read-through annotations and sorted them with bedtools (sort) and extracted the forward and reverse

reads from the bam files with bedtools (bamtofastq). We generated the list of transcript sequences from the genome and the annotation with bedtools (getfasta -s). The transcript sequences were then indexed with kallisto (index-k 31 –make-unique) (*Bray et al., 2016*) and the quantification achieved with kallisto (quant –single -l 363.4 s 85.53354). Quantifications were performed separately on the transcript alone and on the transcript plus read-through annotation.

Quantifications in mutant strains compared to wt were performed using the package DESeq2 (*Love et al., 2014*) with R (v3.4.4). For differential expression analyses, we tested for log2 fold change superior to 0.5 or inferior to $-0.5$. For read-through events, we tested for a log2 fold change superior to 0 compared to the wild type to declare the read-through present in the mutant background. For all analyses, we selected p-values with an FDR $\leq 0.05$. The package ggplot2 (v2.2.1) was used for graphics. The orientation of genes was analysed with R scripts.

### High-resolution tiling arrays

For transcriptome analysis in G1 phase, $\Delta bar1$ cells were grown at 30°C in YEP-R to mid-log phase, collected by filtration, washed with $dH_2O$, and re-suspended at an $OD_{600}$ of 0.3 – 0.4 in YEP-R containing 3 μg/ml α-factor. After 1.5 hr, galactose was added to 2% and, after another 2.5 hr, 100 ml cells of $OD_{600}$ = 0.6 – 0.8 were harvested by centrifugation at room temperature for RNA isolation. For transcriptome analysis during G2 phase, cells were grown at 30°C in YEP-R to mid-log phase, collected by filtration, washed with $dH_2O$, and re-suspended at an $OD_{600}$ of 0.3 – 0.4 in YEP-R containing 3 μg/ml α-factor. After one hour, additional α-factor was added to 3 μg/ml and 30 min later galactose was added to 2%. After another 30 min, cells were collected by filtration, washed with $dH_2O$ and re-suspended in YEP-RG with 3 μg/ml α-factor. Fresh α-factor was added to 3 μg/ml after one hour. After another hour, cells were collected by filtration, washed with $dH_2O$ and re-suspended in YEP-RG with 10 μg/ml nocodazole. 1.5 hr after the release from G1 phase, 100 ml culture with $OD_{600}$ of 1.0 were harvested by centrifugation at room temperature for RNA isolation. Samples were collected for FACScan analysis at the indicated time points.

High-resolution tiling arrays were performed and analysed as described (*Xu et al., 2009*). In brief, RNA isolated from yeast cells was reverse transcribed to cDNA with a mixture of random hexamers and oligo-dT primers, labelled and hybridized to tiled Affymetrix arrays of the budding yeast genome (S288c Genome Chip, http://www-sequence.stanford.edu/s288c/1lq.html). Transcripts that were two-fold or more up- or downregulated are listed in *Supplementary file 5*.

### Accession number

RNA-seq data are accessible from the Gene Expression Omnibus (GEO) database under the accession number GSE112281.

### Antibodies

Antibodies used in this study are listed in *Supplementary file 4*.

### Acknowledgements

We thank Vincent Vanoosthuyse and André Verdel for fruitful discussions, and V Vanoosthuyse for critical reading of the manuscript. We are grateful to François Bachand, Jean-Paul Javerzat, Kim Nasmyth, Chris Norbury, Masayuki Yamamoto and the Yeast Genetic Resource Center of the National BioResource Project–Japan for strains, and to Keith Gull for the TAT1 antibody. We thank the Pôle Scientifique de Modélisation Numérique (PSMN) of ENS-Lyon and the Biocomputing Pole of the LBMC for in-silico analyses. We acknowledge the help of Hélène Polveche and Lorraine Soudade during the initial phase of bioinformatics analyses. We are grateful to support from the EMBL Flow Cytometry Core Facility. Clémence Hocquet is supported by a PhD studentship from the Ministère de l'Enseignement Supérieur et de la Recherche and from the Fondation pour la Recherche Médicale (grant FDT20170437039). Xavier Robellet is supported by a post-doctoral fellowship from the ANR. Research in the Bernard lab is supported by the CNRS, the ANR (grant ANR-15-CE12-0002-01), the Fondation ARC pour la Recherche sur le Cancer (grant PJA 20151203343) and the Comité du Rhône de la Ligue Nationale contre le Cancer. This research in the Marguerat lab was supported by the UK Medical Research Council. Research in the Haering and Steimetz labs is supported by EMBL.

## Additional information

### Funding

| Funder | Grant reference number | Author |
|---|---|---|
| Centre National de la Recherche Scientifique | | Pascal Bernard |
| Agence Nationale de la Recherche | ANR-15-CE12-0002-01 | Xavier Robellet<br>Pascal Bernard |
| Fondation ARC pour la Recherche sur le Cancer | PJA 20151203343 | Pascal Bernard |
| Ligue Régionale Contre le Cancer - comité du Rhône | | Pascal Bernard |
| Medical Research Council | | Xi-Ming Sun<br>Samuel Marguerat |
| European Molecular Biology Laboratory | | Sara Cuylen-Haering<br>Sandra Clauder-Münster<br>Lars Steinmetz<br>Christian H Haering |
| Fondation pour la Recherche Médicale | FDT20170437039 | Clémence Hocquet |

The funders had no role in study design, data collection and interpretation, or the decision to submit the work for publication.

### Author contributions

Clémence Hocquet, Xavier Robellet, Formal analysis, Investigation, Visualization, Methodology; Laurent Modolo, Claire Burny, Software, Formal analysis, Methodology; Xi-Ming Sun, Formal analysis, Visualization, Methodology; Sara Cuylen-Haering, Conceptualization, Investigation, Visualization; Esther Toselli, Sandra Clauder-Münster, Investigation, Methodology; Lars Steinmetz, Resources, Methodology; Christian H Haering, Samuel Marguerat, Conceptualization, Supervision, Funding acquisition, Validation, Writing—review and editing; Pascal Bernard, Conceptualization, Supervision, Funding acquisition, Validation, Writing—original draft, Project administration, Writing—review and editing

### Author ORCIDs

Sara Cuylen-Haering https://orcid.org/0000-0002-1193-4648
Christian H Haering https://orcid.org/0000-0001-8301-1722
Pascal Bernard https://orcid.org/0000-0003-2732-9685

### Decision letter and Author response

Decision letter https://doi.org/10.7554/eLife.38517.027
Author response https://doi.org/10.7554/eLife.38517.028

## Additional files

### Supplementary files

• Supplementary File 1. RNA levels of RNA-exosome and TRAMP components in *cut14-208* mutant cells
DOI: https://doi.org/10.7554/eLife.38517.018

• Supplementary File 2. Yeast strains used in this study
DOI: https://doi.org/10.7554/eLife.38517.019

• Supplementary File 3. Primers and RNA-FISH probes used in this study
DOI: https://doi.org/10.7554/eLife.38517.020

• Supplementary File 4. Antibodies used in this study

DOI: https://doi.org/10.7554/eLife.38517.021

- Supplementary File 5. Microarrays analysis nocodazole-arrested cells

DOI: https://doi.org/10.7554/eLife.38517.022

- Transparent reporting form

DOI: https://doi.org/10.7554/eLife.38517.023

## Data availability

RNA-seq data are accessible from the Gene Expression Omnibus (GEO) database under the accession number GSE112281. Microarrays data are available as supplemental table in excel format

The following dataset was generated:

| Author(s) | Year | Dataset title | Dataset URL | Database, license, and accessibility information |
|---|---|---|---|---|
| Hocquet C, Robellet X, Modolo L, Sun X-M, Burny C, Cuylen-Haering S, Toselli E, Clauder-Münster S, Steinmetz LM, Haering CH, Marguerat S, Bernard P | 2018 | Condensin controls cellular RNA levels through the accurate segregation of chromosomes instead of directly regulating transcription | http://www.ncbi.nlm.nih.gov/geo/query/acc.cgi?acc=GSE112281 | Publicly available at the NCBI Gene Expression Omnibus (accession no: GSE112281). |

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
