## [Decision Letter]

Thank you for submitting your article "Condensin controls cellular RNA levels through the accurate segregation of chromosomes instead of directly regulating transcription" for consideration by *eLife*. Your article has been reviewed by three peer reviewers, including Job Dekker as the Reviewing Editor and Reviewer #1, and the evaluation has been overseen by Kevin Struhl as the Senior Editor.

The reviewers have discussed the reviews with one another and the Reviewing Editor has drafted this decision to help you prepare a revised submission.

Summary:

The manuscript investigates the role of condensin in regulation of gene expression using fission and budding yeast as model systems. A role for condensins in gene expression has been suggested by many studies but the actual mechanism remains to be elucidated. In this study, the authors conducted a series of challenging experiments by depleting condensin in cells and assaying RNA expression at various phases of the cell cycle. This allowed them to make the following conclusions: First, they show that major changes in gene expression are only observed when condensin depleted cells are allowed to complete cytokinesis. They then present evidence that condensin does not play a direct role in regulation of gene expression but rather manifests its effects through problems with accurate chromosome segregation. Finally, they claim that missegregation of rDNA, and thus the unequal distribution of nuclear exosome components within the daughter cells, is the major contributor towards the observed differential expression in fission yeast.

Overall, all three reviewers agreed that the manuscript is well written and experiments are clearly described. The analyses are rigorous and mostly support the major conclusions drawn by the authors. However, there are some issues which need to be addressed:

Essential revisions:

1) The strongest part of the work is the analysis of *S. pombe*, while the analysis of budding yeast was less extensive. The authors should either completely remove all the analysis of budding yeast, or add significant new data to strengthen that part of the manuscript.

2) The authors should better relate their data to previous studies, e.g. tRNA expression changes and comparison of previously published RNA data in *cut3-477* cells.

3) Although the authors do show strong data that formation of anucleolate cell formation in condensin mutants can explain many effects on RNA levels, there are still observations that suggest that other processes may play roles as well. Specifically, a) other chromosome losses (other than loss of the nucleolus) can also occur (e.g. loss of the exosome encoding genes), b) some anucleolate cells do not display altered mug93as transcripts and c) not all condensin-dependent transcript changed depend on the RNA exosome. The authors should address, or at least discuss, these issues.

---

## [Author Response]

Essential revisions:1) The strongest part of the work is the analysis of S. pombe, while the analysis of budding yeast was less extensive. The authors should either completely remove all the analysis of budding yeast, or add significant new data to strengthen that part of the manuscript.

Indeed, the main focus of our work is the analysis of the consequences of condensin inactivation on gene transcription in fission yeast. The key purpose of the budding yeast experiments is to show that the major conclusion that condensin inactivation does not directly affect the expression of the bulk of genes in non-dividing cells, is correct also in an evolutionary distant organism. And this is precisely what the budding yeast experiments unambiguously show – without the necessity to repeat all experiments that we have performed in fission yeast in budding yeast.

Comparing budding and fission yeast in the context of this study is particularly relevant, because it allows to address the important question as to whether the ability (or not) of condensin to regulate gene expression is influenced by the dynamics of its chromosomal localisation during the cell cycle. Indeed condensin is mainly cytoplasmic during interphase and enriched on chromosomes during mitosis in fission yeast, while it is nuclear throughout the cell cycle in budding yeast. These different behaviours are reminiscent of the condensin I and II complexes in mammals. Therefore, showing that condensin plays no major role in the control of gene expression in both fission and budding yeasts rules out the possibility that the results obtained in fission yeast are caused by the specific cell cycle localisation of condensin in this species. Removing the budding yeast part of the work would, in our view, detrimentally weaken the message conveyed by the manuscript.

Nevertheless, we have now performed additional budding yeast experiments to demonstrate missegregation of exosome components also in this species.

2) The authors should better relate their data to previous studies, e.g. tRNA expression changes and comparison of previously published RNA data in cut3-477 cells.

We thank the reviewers for this suggestion. We have now fully addressed this point by analysing the expression level of tRNAs in condensin mutant strains. Please note that tRNA levels cannot be analysed by conventional RNA-seq, presumably because their small, structured and extensively-modified nature impose a dedicated protocol for retrotranscription (see Wilusz, 2015). We therefore used RT-qPCR, which has the advantage of allowing a direct comparison with the previous work of Iwazaki et al., 2010. To further strengthen the comparison, we have analysed 5 reporter tRNAs, amongst which three have been shown to accumulate in *cut3-477* mutant cells by Iwasaki et al. We measured the abundance of these 5 reporter tRNAs in six different condensin mutants, named *cnd1-175, cut3-i23, cut3-477, cut3-m26, cut14-90* and *cut14-208*, of increasing severity, as judged by their frequencies of chromosome missegregation.

We failed to detect any increase in tRNA levels in *cut3-477* cells grown at the restrictive temperature for 2.5 hours, and in that sense did not reproduce the results of Iwasaki et al. Please, note that in their published study, Iwazaki et al. shifted *cut3-477* cells to 36°C for 2 hours. Nevertheless, we observed robust increases of tRNA levels in the *cut3-m26, cut14-90* and *cut14-208* mutant cells. Importantly, we collected evidence that the accumulation of tRNAs in condensin mutant cells is the indirect consequence of chromosome missegregation and aneuploidy. We found (1) that the amplitude of the increase in tRNA levels correlates with the frequency of chromosome missegregation, *cut14-208* being the most severely affected for both phenotypes, (2) that preventing the formation of aneuploid and anucleolate cells by the *cdc15-140* mutation restored normal levels of tRNAs in *cut14-208* mutant cells, despite condensin being impaired, and (3) that non-condensin mutations that disrupt accurate chromosome segregation, such as *ptr4-1, cut9-665* and *mis6-302*, all cause an increase in tRNA levels.

Thus, these new data corroborate our previous findings with RNA Pol II-transcribed mRNAs and ncRNAs. We therefore propose that the imbalance in genomic content that results from chromosome missegregation during mitosis is a major cause of the deregulation of most, if not all, cellular RNAs when condensin function is impaired. The results regarding tRNA levels are presented in the revised version of the manuscript in Figure 6 and Figure 7—figure supplement 1B, and in the last paragraph of the subsection “Gene expression changes in fission yeast condensin mutants are the result of a loss of genome integrity”.

Please note, however, that although tRNA levels were very similar to *cut14-208* in *ptr4-1* and *cut9-665* cells, we observed a difference in the level of two reporter tRNAs in *mis6-302* compared to *cut14-208*. This observation is consistent with the existence of gene-specific effects associated with the gain or loss of chromosomes (for reviews, see Pfau and Amon, 2012; Santaguida and Amon, 2015). It might also suggest that different variations of chromosome missegregation might have slightly different impacts on the whole transcriptome. This finding brought us to slightly revise the part of our Discussion section on the impact of aneuploidy on the transcriptome of condensin mutant cells.

3) Although the authors do show strong data that formation of anucleolate cell formation in condensin mutants can explain many effects on RNA levels, there are still observations that suggest that other processes may play roles as well. Specifically, a) other chromosome losses (other than loss of the nucleolus) can also occur (e.g. loss of the exosome encoding genes), b) some anucleolate cells do not display altered mug93as transcripts and c) not all condensin-dependent transcript changed depend on the RNA exosome. The authors should address, or at least discuss, these issues.

a) We agree that the loss of chromosomal fragments other than the rDNA/nucleolus most likely contribute to changes in the transcriptome of condensin mutant cells. This is what we have indicated in the first version of the manuscript. We have rephrased the Discussion to make this point clearer.

Regarding the loss of exosome-encoding genes, although this is formally possible, we would like to emphasize that, as we had mentioned in the first version of the manuscript (and Supplementary file 1), we did not detect any reduction in the mRNA levels neither of the exosome components, nor of the TRAMP components, at least at the level of the whole population of *cut14-208* cells. This is now indicated in the last paragraph of the subsection “Read-through transcripts accumulate upon condensin inactivation” (Supplementary file 1). Furthermore, given that *cut14-208* mutant cells were allowed to go through a single cell cycle during the relatively short 2.5 h time-window of the experiment, we consider very unlikely that the loss of exosome-encoding genes might account for the changes in RNA levels that we measured in the whole population of cells by RNA-seq.

b) and c). We thank the reviewers for this comment. These two observations can be explained by the fact that both Rrp6 and Dis3 are reduced, but not totally eliminated from anucleolate *cut14-208* cells. The persistence of traces of Rrp6 and of Dis3 in anucleolate *cut14-208* cells can be clearly observed in Figure 7B (Rrp6) and Figure 7—figure supplement 1D (Dis3). This result implies that a fraction of the Rrp6 and Dis3 molecules is transmitted to daughter cells independently of the nucleolus, presumably in association with the bulk of chromatin, or within the nucleoplasm. The presence of residual Rrp6 and Dis3 molecules in anucleolate *cut14-208* cells provides a straightforward explanation to the fact that anucleolate cells with low chromatin amount, as judged by DAPI staining, exhibit the strongest mug93as FISH signals, and conversely why other anucleolate cells with high chromatin amounts show weak or no mug93as FISH signals. Note that the missegregation of the fragment of chromosome 2 that bears the *mug93* gene can also explain why some anucleolate cells do not exhibit a mug93as FISH signal.

The partial reduction of both Rrp6 and Dis3 in *cut14-208* mutant cells also explains why the transcriptomes of *cut14-208* and *rrp6Δ* cells overlap only partially, since only the RNA molecules that are the most sensitive to the dosage of Rrp6 (and to the dosage of Dis3), will accumulate in *cut14-208*. Other RNAs that are less sensitive to the reduction of Rrp6 will not accumulate in *cut14-208* cells, in contrast to *rrp6Δ* cells.

We agree this is an important point that needs to be made crystal clear. We now clearly mention this point in the second paragraph of the Results subsection “Non-disjunction of the rDNA during anaphase depletes the RNA-exosome from post-mitotic fission yeast cells and results in ncRNA accumulation”, and also in the fourth paragraph of the Discussion section.